# Optical estimation of absolute membrane potential using fluorescence lifetime imaging

Julia R Lazzari-Dean[1], Anneliese MM Gest[1], Evan W Miller[1,2,3]*

[1]Department of Chemistry, University of California, Berkeley, Berkeley, United States; [2]Department of Molecular & Cell Biology, University of California, Berkeley, Berkeley, United States; [3]Helen Wills Neuroscience Institute, University of California, Berkeley, Berkeley, United States

**Abstract** All cells maintain ionic gradients across their plasma membranes, producing transmembrane potentials ($V_{mem}$). Mounting evidence suggests a relationship between resting $V_{mem}$ and the physiology of non-excitable cells with implications in diverse areas, including cancer, cellular differentiation, and body patterning. A lack of non-invasive methods to record absolute $V_{mem}$ limits our understanding of this fundamental signal. To address this need, we developed a fluorescence lifetime-based approach (VF-FLIM) to visualize and optically quantify $V_{mem}$ with single-cell resolution in mammalian cell culture. Using VF-FLIM, we report $V_{mem}$ distributions over thousands of cells, a 100-fold improvement relative to electrophysiological approaches. In human carcinoma cells, we visualize the voltage response to growth factor stimulation, stably recording a 10–15 mV hyperpolarization over minutes. Using pharmacological inhibitors, we identify the source of the hyperpolarization as the $Ca^{2+}$-activated $K^+$ channel $K_{Ca}3.1$. The ability to optically quantify absolute $V_{mem}$ with cellular resolution will allow a re-examination of its signaling roles.
DOI: https://doi.org/10.7554/eLife.44522.001

## Introduction

Membrane potential ($V_{mem}$) is an essential facet of cellular physiology. In electrically excitable cells, such as neurons and cardiomyocytes, voltage-gated ion channels enable rapid changes in membrane potential. These fast membrane potential changes, on the order of milliseconds to seconds, trigger release of neurotransmitters in neurons or contraction in myocytes. The resting membrane potentials of these cells, which change over longer timescales, affect their excitability. In non-electrically excitable cells, slower changes in $V_{mem}$—on the order of seconds to hours—are linked to a variety of fundamental cellular processes (*Abdul Kadir et al., 2018*), including mitosis (*Cone and Cone, 1976*), cell cycle progression (*Huang and Jan, 2014*), and differentiation (*Tsuchiya and Okada, 1982*). Mounting lines of evidence point to the importance of electrochemical gradients in development, body patterning, and regeneration (*Levin, 2014*).

Despite the importance of membrane potential to diverse processes over a range of time scales, the existing methods for recording $V_{mem}$ are inadequate for characterizing distributions of $V_{mem}$ states in a sample or studying gradual shifts in resting membrane potential (*Figure 1—source data 1*). Patch clamp electrophysiology remains the gold standard for recording cellular electrical parameters, but it is low throughput, highly invasive, and difficult to implement over extended time periods. Where reduced invasiveness or higher throughput analyses of $V_{mem}$ are required, optical methods for detecting events involving $V_{mem}$ changes (e.g. whether an action potential occurred) are often employed (*Huang et al., 2006*; *McKeithan et al., 2017*; *Zhang et al., 2016*). However, optical approaches generally use fluorescence intensity values as a readout, which cannot report either the

*For correspondence:
evanwmiller@berkeley.edu

**eLife digest** All living cells are like tiny batteries. As long as a cell is alive, it actively maintains a difference in electrical charge between its interior and exterior. This charge difference, or voltage, is called the membrane potential, and it is vital for our bodies to work properly. For example, fast changes in membrane potential control our heartbeat and underpin the electrical signals that brain cells use to communicate.

Slower changes in membrane potential – ranging from minutes to days – may also play important roles in other organs. To understand how and why membrane potential is important in these contexts, we need methods to measure it accurately in individual cells.

One way is to puncture cells with microscopic electrodes: this yields accurate results but damages the cells and can only measure one cell at a time. Alternative methods treat cells with special fluorescent dyes and then image them with a microscope. The dyes emit light in response to voltage variations: when the cells' membrane potential changes, the dyes glow brighter. The changes in light intensity give an estimate of the size of the change in membrane potential. This allows many cells to be analyzed without harming them, but it is less accurate.

Fluorescence lifetime refers to how long fluorescent dyes take to finish emitting light, and this phenomenon has already helped researchers to record a variety of processes in the cell. Lazzari-Dean et al. therefore wanted to use fluorescence lifetime to develop a better way of recording membrane potential. This method, called VF-FLIM, relied on measuring how long certain dyes took to finish emitting light at specific voltages, rather than how bright they were.

Experiments using mammalian cells grown in the laboratory showed that the membrane potentials measured with VF-FLIM were similar to those recorded with electrodes, which represent the highest standard of accuracy. The new method was at least eight times more accurate than other techniques using fluorescent dyes. VF-FLIM could also measure many thousands of cells within a few hours, a hundred times faster than electrode-based methods. Finally, tests on human cancer cells revealed that VF-FLIM could detect that these cells go through gradual changes in membrane potential in response to growth signals.

VF-FLIM is a new, non-invasive tool that can measure changes in membrane potential more quickly and accurately. This will help to better understand the many roles membrane potential could play in healthy and diseased cells.

DOI: https://doi.org/10.7554/eLife.44522.002

value of $V_{mem}$ in millivolts ('absolute $V_{mem}$') or the millivolt amount by which $V_{mem}$ changed (*Peterka et al., 2011*). Variations in dye environment (*Ross and Reichardt, 1979*), dye loading, illumination intensity, fluorophore bleaching, and/or cellular morphology complicate fluorescence intensity measurements, making calibration and determination of absolute membrane potential difficult or impossible. This limitation restricts optical analysis to detection of acute $V_{mem}$ changes, which can be analyzed without comparisons of $V_{mem}$ between cells or over long timescales.

One strategy to address these fluorescence intensity artifacts and quantify cellular parameters optically is ratio-based imaging. For $V_{mem}$ specifically, ratio-based signals can be accessed either with a two-component system or with an electrochromic voltage sensitive dye, but neither strategy has enabled accurate absolute $V_{mem}$ recordings. Two-component FRET-oxonol systems, with independent chromophores for ratio-based calibration, have seen limited success (*González and Tsien, 1997*), and they confer significant capacitive load on the cell (*Briggman et al., 2010*). Further, their performance hinges on carefully tuned loading procedures of multiple lipophilic indicators (*Adams and Levin, 2012*), which can be challenging to reproduce across different samples and days. On the other hand, electrochromic probes report voltage as changes in excitation and emission wavelengths of a single chromophore (*Loew et al., 1979*). While they benefit from simpler loading procedures, signals from electrochromic styryl dyes require normalization with an electrode on each cell of interest to determine absolute $V_{mem}$ accurately (*Montana et al., 1989*; *Zhang et al., 1998*; *Bullen and Saggau, 1999*). As a result, ratiometric $V_{mem}$ sensors cannot be used to optically quantify slow signals in the resting $V_{mem}$, which may be on the order of tens of millivolts. Indeed,

ratiometric $V_{mem}$ probes are most commonly applied to detect - rather than quantify - fast changes in $V_{mem}$ (*Zhang et al., 1998*), much like their single wavelength counterparts.

An alternative approach to improved quantification in optical measurements is fluorescence lifetime ($\tau_{fl}$) imaging (FLIM), which measures the excited state lifetime of a population of fluorophores. Because fluorescence lifetime is an intrinsic property, FLIM can avoid many of the artifacts that confound extrinsic fluorescence intensity measurements, such as uneven dye loading, fluorophore bleaching, variations in illumination intensity, and detector sensitivity (*Berezin and Achilefu, 2010*; *Yellen and Mongeon, 2015*). If a fluorescent probe responds to the analyte of interest via changes in the lifetime of its excited state, there is the opportunity to use fluorescence lifetime to provide a more quantitative estimate of analyte parameters than can be achieved with fluorescence intensity alone. Although FLIM measurements can be affected by environmental factors such as temperature, ionic strength and local environment (*Berezin and Achilefu, 2010*), FLIM has been widely employed to record a number of biochemical and biophysical parameters, including intracellular $Ca^{2+}$ concentration (*Zheng et al., 2015*), viscosity (*Levitt et al., 2009*), GTPase activity (*Harvey et al., 2008*), kinase activity (*Lee et al., 2009*), and redox state (NADH/NAD$^+$ ratio) (*Blacker and Duchen, 2016*), among others (*Yellen and Mongeon, 2015*). Attempts to record absolute voltage with FLIM, however, have been limited in success (*Dumas and Stoltz, 2005*; *Hou et al., 2014*; *Brinks et al., 2015*). Previous work focused on genetically encoded voltage indicators (GEVIs), which either possess complex relationships between $\tau_{fl}$ and voltage (*Hou et al., 2014*) or show low sensitivity to voltage in lifetime (*Brinks et al., 2015*) and require complex and technically challenging measurements of fast photochemical kinetics to estimate voltage (*Hou et al., 2014*). Because of this poor voltage resolution, the fluorescence lifetimes of GEVIs cannot be used to detect most biologically relevant voltage changes, which are on the order of tens of millivolts.

Fluorescent voltage indicators that use photoinduced electron transfer (PeT) as a voltage-sensing mechanism are promising candidates for a FLIM-based approach to optical $V_{mem}$ quantification. Because PeT affects the nonradiative decay rate of the fluorophore excited state, it has been successfully translated from intensity to $\tau_{fl}$ imaging with a number of small molecule probes for $Ca^{2+}$ (*Lakowicz et al., 1992*). We previously established that VoltageFluor (VF)-type dyes transduce changes in cellular membrane potential to changes in fluorescence intensity and that the voltage response of VF dyes is consistent with a PeT-based response mechanism (*Miller et al., 2012*; *Woodford et al., 2015*). Changes in the transmembrane potential alter the rate of PeT (*Li, 2007*; *de Silva et al., 1995*) from an electron-rich aniline donor to a fluorescent reporter, thereby modulating the fluorescence intensity of VF dyes (*Miller et al., 2012*) (*Figure 1A,B*). VoltageFluors also display low toxicity and rapid, linear responses to voltage.

Here, we develop fluorescence lifetime imaging of VoltageFluor dyes (VF-FLIM) as a quantitative, all-optical approach for recording absolute membrane potential with single cell resolution. Using patch-clamp electrophysiology as a standard, we demonstrate that VF-FLIM reports absolute membrane potential in single trials with 10 to 23 mV accuracy (root mean square deviation, RMSD; 15 s acquisition), depending on the cell line. In all cases tested, VF-FLIM tracks membrane potential *changes* with better than 5 mV accuracy (RMSD). We benchmark VF-FLIM against previously reported optical absolute $V_{mem}$ recording approaches and demonstrate resolution improvements of 8-fold over ratiometric strategies and 19-fold over other lifetime-based strategies. To highlight the increased throughput relative to manual patch-clamp electrophysiology, we document resting membrane potentials of thousands of cells. To our knowledge, this work represents the first broad view of the distribution of resting membrane potentials present in situ. VF-FLIM is limited to acquisition speeds on the order of seconds, but it is well-suited for studying gradual $V_{mem}$ dynamics. Using VF-FLIM, we quantify and track the evolution of a 10–15 mV $V_{mem}$ hyperpolarization over minutes following epidermal growth factor (EGF) stimulation of human carcinoma cells. Through pharmacological perturbations, we conclude that the voltage changes following EGF stimulation arise from activation of the calcium-activated potassium channel $K_{Ca}3.1$. Our results show that fluorescence lifetime of VF dyes is a generalizable and effective approach for studying resting membrane potential in a range of cell lines (*Lakowicz et al., 1992*).

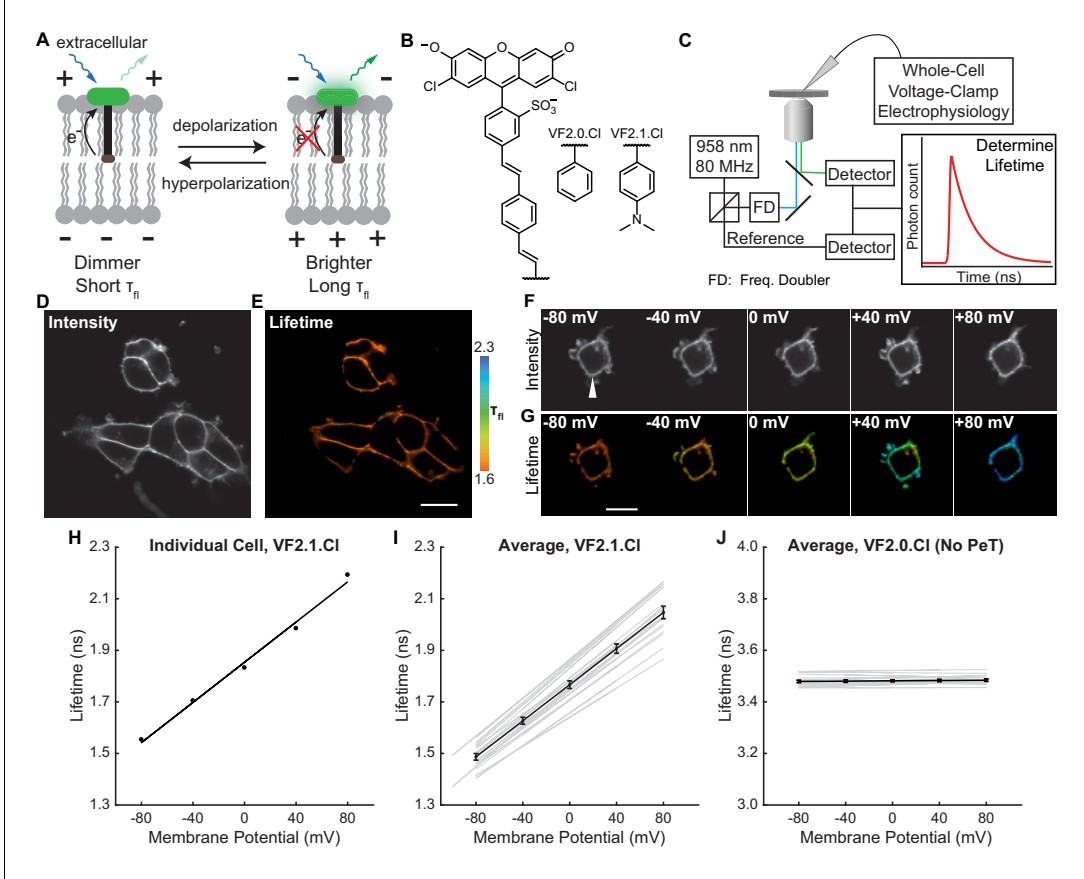

**Figure 1.** VoltageFluor FLIM linearly reports absolute membrane potential. (**A**) Mechanism of VoltageFluor dyes, in which depolarization of the membrane potential attenuates the rate of photoinduced electron transfer. (**B**) Structures of the VF molecules used in this study. (**C**) Schematic of the TCSPC system used to measure fluorescence lifetime. Simultaneous electrophysiology was used to establish lifetime-voltage relationships. (**D**) Fluorescence intensity and (**E**) lifetime of HEK293T cells loaded with 100 nM VF2.1.Cl. (**F**) Intensity and (**G**) lifetime images of HEK293T cells voltage clamped at the indicated membrane potential. (**H**) Quantification of the single trial shown in (**G**), with a linear fit to the data. (**I**) Evaluation of VF2.1.Cl lifetime-voltage relationships in many individual HEK293T cells. Gray lines represent linear fits on individual cells. Black line is the average lifetime-voltage relationship across all cells (n = 17). (**J**) VF2.0.Cl lifetime does not exhibit voltage-dependent changes. Gray lines represent linear fits on individual cells, and the black line is the average lifetime-voltage relationship across all cells (n = 17). Scale bars represent 20 μm. Error bars represent mean ± SEM.

DOI: https://doi.org/10.7554/eLife.44522.003

The following source data and figure supplements are available for figure 1:

**Source data 1.** Comparison of available approaches for measuring membrane potential in cells.
DOI: https://doi.org/10.7554/eLife.44522.008

**Source data 2.** Properties of lifetime standards and VoltageFluor dyes.
DOI: https://doi.org/10.7554/eLife.44522.009

**Source data 3.** Comparison of optical approaches to absolute $V_{mem}$ determination in HEK293T cells.
DOI: https://doi.org/10.7554/eLife.44522.010

**Figure supplement 1.** Overview of data processing to obtain membrane potential recordings from fluorescence lifetime.
DOI: https://doi.org/10.7554/eLife.44522.004

**Figure supplement 2.** Concentration dependence of VoltageFluor lifetimes in HEK293T cells.
DOI: https://doi.org/10.7554/eLife.44522.005

**Figure supplement 3.** VF2.0.Cl lifetime does not depend on membrane potential.
DOI: https://doi.org/10.7554/eLife.44522.011

**Figure supplement 4.** The GEVI CAESR shows variable lifetime-voltage relationships.
DOI: https://doi.org/10.7554/eLife.44522.006

**Figure supplement 5.** Ratiometric $V_{mem}$ determinations with Di-8-ANEPPS in HEK293T cells.
DOI: https://doi.org/10.7554/eLife.44522.007

## Results

### VoltageFluor fluorescence lifetime varies linearly with membrane potential

To characterize how the photoinduced electron transfer process affects fluorescence lifetime, we compared the $\tau_{fl}$ of the voltage-sensitive dye VF2.1.Cl with its voltage-insensitive counterpart VF2.0.Cl (*Figure 1B*). We recorded the $\tau_{fl}$ of bath-applied VF dyes in HEK293T cells using time-correlated single-photon counting (TCSPC) FLIM (*Figure 1C–E*, *Scheme 1*). VF2.1.Cl is localized to the plasma membrane and exhibits a biexponential $\tau_{fl}$ decay with decay constants of approximately 0.9 and 2.6 ns (*Figure 1—figure supplement 1*). For all subsequent analysis of VF2.1.Cl lifetime, we refer to the weighted average $\tau_{fl}$, which is approximately 1.6 ns in HEK293T cell membranes at rest. VF2.0.Cl (*Figure 1B*), which lacks the aniline substitution and is therefore voltage-insensitive (*Woodford et al., 2015*), shows a $\tau_{fl}$ of 3.5 ns in cell membranes, which is similar to the lifetime of an unsubstituted fluorescein (*Magde et al., 1999*) (*Figure 1—source data 2*). We also examined VoltageFluor lifetimes at a variety of dye loading concentrations to test for concentration-dependent changes in dye lifetime, which have been reported for fluorescein derivatives (*Chen and Knutson, 1988*). Shortened VF lifetimes were observed at high dye concentrations (*Figure 1—figure supplement 2*); all subsequent VF-FLIM studies were conducted at dye concentrations low enough to avoid this concentration-dependent change in lifetime.

To assess the voltage dependence of VoltageFluor $\tau_{fl}$, we controlled the plasma membrane potential of HEK293T cells with whole-cell voltage-clamp electrophysiology while simultaneously measuring the $\tau_{fl}$ of VF2.1.Cl (*Figure 1C*). Single-cell recordings show a linear $\tau_{fl}$ response to applied voltage steps, and individual measurements deviate minimally from the linear fit (*Figure 1F–H*). VF2.1.Cl $\tau_{fl}$ is reproducible across different cells at the same resting membrane potential, allowing determination of $V_{mem}$ from $\tau_{fl}$ images taken without concurrent electrophysiology (*Figure 1I*). Voltage-insensitive VF2.0.Cl shows no $\tau_{fl}$ change in response to voltage (*Figure 1J*, *Figure 1—figure supplement 3*), consistent with a $\tau_{fl}$ change in VF2.1.Cl arising from a voltage-dependent PeT process. In HEK293T cells, VF2.1.Cl exhibits a sensitivity of $3.50 \pm 0.08$ ps/mV and a 0 mV lifetime of $1.77 \pm 0.02$ ns, corresponding to a fractional change in $\tau_{fl}$ ($\Delta\tau/\tau$) of $22.4 \pm 0.4\%$ per 100 mV. These values are in good agreement with the 27% ΔF/F intensity change per 100 mV originally observed for VF2.1.Cl (*Miller et al., 2012*; *Woodford et al., 2015*). Because %ΔF/F is a fluorescence intensity-based metric, it cannot be used to measure absolute $V_{mem}$; however, agreement between %ΔF/F and %Δτ/τ is consistent with a PeT-based $V_{mem}$ sensing mechanism in VFs. To estimate the voltage resolution of VF-FLIM, we analyzed the variability in successive measurements on the same cell (intra-cell resolution) and on different cells (inter-cell resolution, see Materials and methods). We estimate that the resolution for tracking and quantifying voltage changes in a single HEK293T cell is $3.5 \pm 0.4$ mV (intra-cell resolution, average RMSD from each electrophysiological calibration, Scheme 2), whereas the resolution for single-trial determination of a particular HEK293T cell's absolute $V_{mem}$ is 19 mV (inter-cell resolution, RMSD of each calibration slope to the average calibration, Scheme 2) within a 15 s bandwidth.

We compared the performance of VF-FLIM in HEK293T cells to that of two previously documented strategies for optical absolute $V_{mem}$ determination. We first tested the voltage resolution of CAESR, the best previously reported GEVI for recording absolute $V_{mem}$ with FLIM (*Brinks et al., 2015*). Using simultaneous FLIM and voltage-clamp electrophysiology, we determined the relationship between $\tau_{fl}$ and $V_{mem}$ for CAESR under one photon excitation (*Figure 1—figure supplement 4*). We recorded a sensitivity of $-1.2 \pm 0.1$ ps/mV and a 0 mV lifetime of $2.0 \pm 0.2$ ns, which corresponds to a $-6.1 \pm 0.8\%$ $\Delta\tau/\tau$ per 100 mV (mean ± SEM of 9 measurements), in agreement with the reported sensitivity of $-0.9$ ps/mV and 0 mV lifetime of 2.7 ns with 2 photon excitation (*Brinks et al., 2015*). Relative to VF2.1.Cl, CAESR displays 3-fold lower sensitivity ($-1.2$ ps/mV vs 3.5 ps/mV in HEK293T cells) and 7-fold higher voltage-independent variability in lifetime (0.46 ns vs 0.07 ns, standard deviation of the 0 mV lifetime measurement). For CAESR in HEK293T cells, we calculate a voltage resolution of $33 \pm 7$ mV for quantifying voltage changes on an individual cell (intra-cell RMSD, compared to 3.5 mV for VF2.1.Cl, see Materials and methods) and resolution of 370 mV for determination of a particular cell's absolute $V_{mem}$ (inter-cell RMSD, compared to 19 mV for VF2.1.Cl).

We also measured the absolute voltage resolution of the ratio-based sensor di-8-ANEPPS, which reports membrane potential by the wavelength of its excitation and emission spectra (*Loew et al., 1979*). Ratio-based imaging can be achieved by comparing the fluorescence emission at different excitation wavelengths (*Zhang et al., 1998*); here, we used the ratio, R, of the blue-excited emission to the green-excited emission (see Materials and methods). Via simultaneous ratio imaging and whole cell voltage clamp electrophysiology, we record a sensitivity of 0.0039 ± 0.0004 R per mV, with a y-intercept (0 mV) R value of 1.8 ± 0.2 (*Figure 1—figure supplement 5*; mean ± SEM of n = 16 HEK293T cells). R depends on the excitation and emission conditions used but should be relatively reproducible on a given microscope rig. To compare R from our system with previous work, we normalized all R values to the R value at 0 mV for each cell. Using the above data, we obtain a sensitivity of 0.0022 ± 0.0002 normalized R per mV, with a 0 mV normalized R of 1.02 ± 0.02, in good agreement with reported values (0.0015 normalized R per mV) (*Zhang et al., 1998*). For analysis of voltage resolution, we compare VF-FLIM to the non-normalized R, since normalization requires an electrode-based measurement for every recording and is thus not a truly optical strategy. From the non-normalized di-8-ANEPPS R, we obtain an intra-cell resolution (RMSD) of 18 ± 3 mV (5-fold less accurate than VF-FLIM) and an inter-cell resolution (RMSD) of 150 mV (8-fold less accurate than VF-FLIM). The sensitivities and resolutions of VF-FLIM, CAESR, and di-8-ANEPPS in HEK293T are tabulated in *Figure 1—source data 3*. Because cellular resting membrane potentials and voltage changes (e.g. action potentials) are on the order of tens of millivolts, the resolution improvements achieved by VF-FLIM enable biologically relevant absolute $V_{mem}$ recordings: impossible with previous approaches.

## Evaluation of VF-FLIM across cell lines and culture conditions

To test the generalizability of VF-FLIM, we determined $\tau_{fl}$-$V_{mem}$ calibrations in four additional commonly used cell lines: A431, CHO, MDA-MB-231, and MCF-7 (*Figure 2*, *Figure 2—figure supplement 1*, *Figure 2—figure supplement 2*). We observe a linear $\tau_{fl}$ response in all cell lines tested. The slope (voltage sensitivity) and y-intercept (0 mV lifetime) of the $\tau_{fl}$-$V_{mem}$ response varied slightly across cell lines, with average sensitivities of 3.1 to 3.7 ps/mV and average 0 mV lifetimes ranging from 1.74 to 1.87 ns. In all cell lines, we observed better voltage resolution for quantification of $V_{mem}$ changes on a given cell versus comparisons of absolute $V_{mem}$ between cells. Changes in voltage for a given cell could be quantified with resolutions at or better than 5 mV (intra-cell resolution, Materials and methods). For absolute $V_{mem}$ determination of a single cell, we observed voltage resolutions ranging from 10 to 23 mV (inter-cell resolution, 15 s acquisition time, *Figure 2—source data 1*). Statistically significant differences among the cell lines tested were observed for cellular $\tau_{fl}$-$V_{mem}$ calibrations in both the slope (One-way ANOVA with Welch's correction: $F_{(4, 23.07)}$=18.12, p<0.0001) and average 0 mV lifetime (One-way ANOVA: $F_{(4, 67)}$=14.43, p<0.0001). There were no statistically significant differences between A431, CHO, and HEK293T cells (p>0.05, Games-Howell and Tukey-Kramer post hoc tests for the slope and 0 mV lifetime respectively). MDA-MB-231 and MCF-7 cells showed statistically significant variability from other cell lines in slope and/or 0 mV lifetime.

To verify that VF-FLIM was robust in groups of cells in addition to the isolated, single cells generally used for patch clamp electrophysiology, we determined lifetime-voltage relationships for small groups of A431 cells (*Figure 2—figure supplement 3A–E*). We found that calibrations made in small groups of cells are nearly identical to those obtained on individual cells, indicating that VF-FLIM only needs to be calibrated once for a given type of cell. For pairs or groups of three cells we recorded a sensitivity of 3.3 ± 0.2 ps/mV and a 0 mV lifetime of 1.78 ± 0.02 ns (mean ± SEM of 7 cells (5 pairs and 2 groups of 3); values are for the entire group, not just the cell in contact with the electrode), which is similar to the sensitivity of 3.55 ± 0.08 ps/mV and 0 mV lifetime of 1.74 ± 0.02 ns we observe in single A431 cells. The slight reduction in sensitivity seen in cell groups is likely attributable to space clamp error, which prevents complete voltage clamp of the cell group (*Williams and Mitchell, 2008*; *Armstrong and Gilly, 1992*). Indeed, when we analyzed only the most responsive cell in the group (in contact with the electrode), we obtained a slope of 3.7 ± 0.1 ps/mV and 0 mV lifetime of 1.79 ± 0.02 ns, in good agreement with the single cell data. The space clamp error can be clearly visualized in *Figure 2—figure supplement 3E*, where one cell in the group of 3 responded much less to the voltage command.

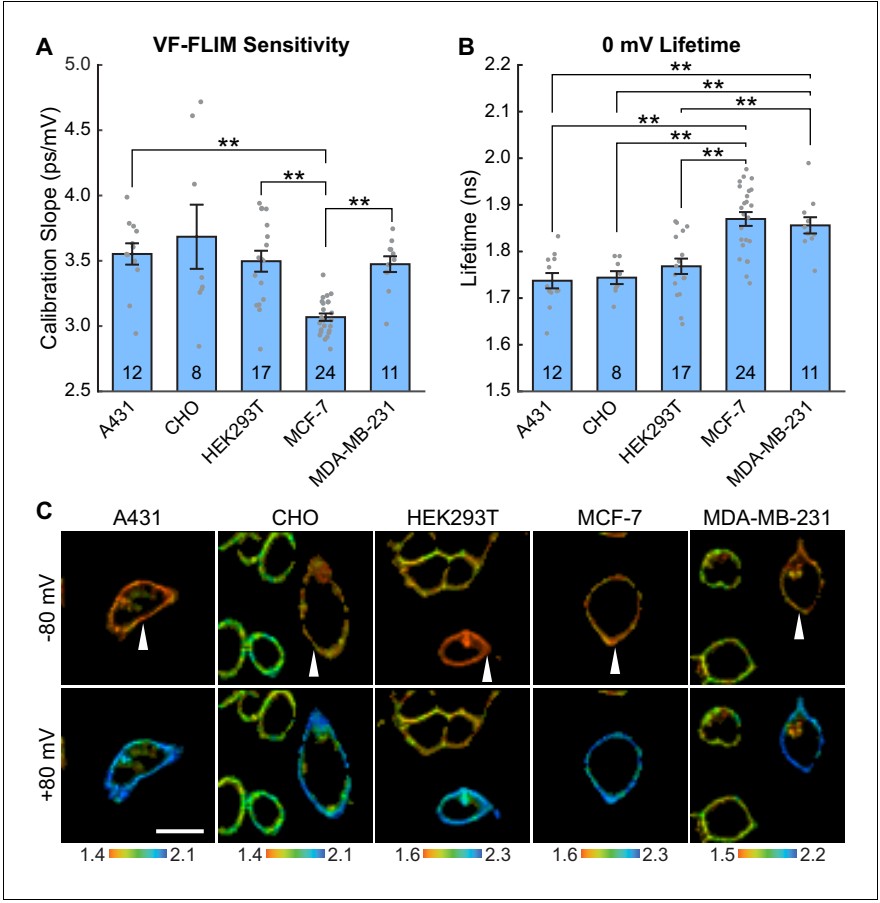

**Figure 2.** VF-FLIM is a general and portable method for optically determining membrane potential. VF2.1.Cl lifetime-voltage relationships were determined with whole cell voltage clamp electrophysiology in five cell lines. (**A**) Slopes of the linear fits for single cell lifetime-voltage relationships, shown as mean ± S.E.M. Gray dots indicate results from individual cells. Statistically significant differences exist between groups (One-way ANOVA with Welch's correction: $F_{(4, 23.07)}=18.12$, $p<0.0001$). Data were tested for normality (Shapiro-Wilk test, $p>0.05$ for all cell lines) and homoscedasticity (Levene's test on the median, $F_{(4,67)} = 5.07$, $p=0.0013$). ** indicates $p<0.01$; if significance is not indicated, $p>0.05$ (Games-Howell post hoc test). (**B**) 0 mV reference point of linear fits for the lifetime-voltage relationship, shown as mean ± S.E.M. Gray dots indicate results from individual cells. Significant differences exist between groups (One-way ANOVA: $F_{(4, 67)}=14.43$, $p<0.0001$). Data were tested for normality (Shapiro-Wilk test, $p>0.05$ for all cell lines) and homoscedasticity (Levene's test on the median, $F_{(4,67)} = 1.29$, $p=0.28$). ** indicates $p<0.01$; if significance is not indicated, $p>0.05$ (Tukey-Kramer post hoc test). (**C**) Representative lifetime-intensity overlay images for each cell line with the indicated cells (white arrow) held at −80 mV (top) or +80 mV (bottom). Lifetime scales are in ns. Scale bar is 20 µm.

DOI: https://doi.org/10.7554/eLife.44522.013

The following source data and figure supplements are available for figure 2:

**Source data 1.** Lifetime-$V_{mem}$ standard curves for VF2.1.Cl lifetime in various cell lines.
DOI: https://doi.org/10.7554/eLife.44522.018

**Figure supplement 1.** VoltageFluor lifetime reports voltage in diverse cell lines.
DOI: https://doi.org/10.7554/eLife.44522.014

**Figure supplement 2.** Additional parameters of linear lifetime-voltage standard curves.
DOI: https://doi.org/10.7554/eLife.44522.015

**Figure supplement 3.** Relationship between lifetime and membrane potential extends to groups of cells and across culture conditions.
DOI: https://doi.org/10.7554/eLife.44522.016

**Figure supplement 4.** Concentration dependence of VoltageFluor lifetime in four cell lines.
DOI: https://doi.org/10.7554/eLife.44522.017

To test whether VF-FLIM is also extensible to cells maintained with different culture conditions, we recorded lifetime-$V_{mem}$ relationship in serum-starved A431 cells (*Figure 2—figure supplement 3F–K*), obtaining an average sensitivity of 3.6 ± 0.1 ps/mV and a 0 mV lifetime of 1.76 ± 0.01 ns (n = 7; two single cells, two pairs, 3 groups of 3 cells; values are average lifetime across the whole cell group), in excellent agreement with the values obtained for non-serum starved cells. We also tested for concentration-dependent changes in VF lifetime in all five cell lines and in serum starvation conditions. Similar to VF2.1.Cl lifetime in HEK293T cells (*Figure 1—figure supplement 2*), we observed shortening of VF2.1.Cl lifetimes beginning between 200 and 500 nM dye in all cases (*Figure 2—figure supplement 4*). All subsequent experiments were carried out at VF2.1.Cl concentrations well below the regime where VF concentration-dependent lifetime changes were observed.

## Optical determination of resting membrane potential distributions

The throughput of VF-FLIM enables cataloging of resting membrane potentials of thousands of cells in only a few hours of the experimenter's time. We optically recorded resting membrane potential distributions for A431, CHO, HEK293T, MCF-7, and MDA-MB-231 cells using VF-FLIM (*Figure 3*, *Figure 3—figure supplement 1*, *Figure 3—figure supplement 2*). We report resting membrane potentials by cell group (Materials and methods, *Figure 1—figure supplement 1*) because adjacent cells in these cultures are electrically coupled to some degree via gap junctions (*Meşe et al., 2007*). Each group of cells represents an independent sample for $V_{mem}$. In addition, the fluorescent signal originating from membranes of adjacent cells cannot be separated with a conventional optical microscope, so assignment of a region of membrane connecting multiple cells would be arbitrary. VF-FLIM images (*Figure 3*, *Figure 3—figure supplement 1*, *Figure 3—figure supplement 2*) contain spatially resolved voltage information, but caution should be employed in interpreting pixel to pixel differences in lifetime. Because VF-FLIM was calibrated here using the average plasma membrane $\tau_{fl}$ for each cell, optical $V_{mem}$ should be interpreted per cell or cell group.

Mean resting membrane potentials recorded by VF-FLIM range from −53 to −29 mV, depending on the cell line. These average $V_{mem}$ values fall within the range reported in the literature for all of the cell lines we measured (*Figure 3—source data 1*). We also recorded resting membrane potentials in a high K$^+$ buffer (120 mM K$^+$, 'high K$^+$ HBSS'), where we observed a depolarization of 15 to 41 mV, bringing the mean $V_{mem}$ up to −26 mV to +4 mV, again depending on the cell line. Although 120 mM extracellular K$^+$ should be strongly depolarizing, it will not necessarily produce a membrane potential of 0 mV. Because few literature reports of electrophysiological measurements in 120 mM K$^+$ exist as a point of comparison, we obtained a rough estimate of $V_{mem}$ in 6 mM extracellular K$^+$ and 120 mM extracellular K$^+$ using the Goldman-Hodgkin-Katz (GHK) equation (*Hodgkin and Katz, 1949*). Under our imaging conditions and with a broad range of possible ion permeabilities and intracellular ion concentrations, the GHK equation allows $V_{mem}$ ranging from −91 to −27 mV in 6 mM extracellular K$^+$ and −25 to +2 mV in 120 mM extracellular K$^+$ (see Materials and methods). Recorded VF-FLIM values fall well within this allowed range. Notably, although the GHK equation can determine ranges of reasonable $V_{mem}$ values, GHK-based $V_{mem}$ results are approximate at best because of the difficulty in obtaining accurate values of permeabilities and intracellular ion concentrations for specific cell lines. Direct measurement of $V_{mem}$, rather than theoretical calculation, is required to obtain accurate values.

## Membrane potential dynamics in epidermal growth factor signaling

We thought VF-FLIM was a promising method for elucidating the roles of membrane potential in non-excitable cell signaling. Specifically, we wondered whether VF-FLIM might be well-suited to dissect conflicting reports surrounding changes in membrane potential during EGF/EGF receptor (EGFR)-mediated signaling. Receptor tyrosine kinase (RTK)-mediated signaling is a canonical signaling paradigm for eukaryotic cells, transducing extracellular signals into changes in cellular state. Although the involvement of second messengers like Ca$^{2+}$, cyclic nucleotides, and lipids are well characterized, membrane potential dynamics and their associated roles in non-excitable cell signaling remain less well-defined. In particular, the activation of EGFR via EGF has variously been reported to be depolarizing (*Rothenberg et al., 1982*), hyperpolarizing (*Pandiella et al., 1989*), or electrically silent (*Moolenaar et al., 1982*; *Moolenaar et al., 1986*).

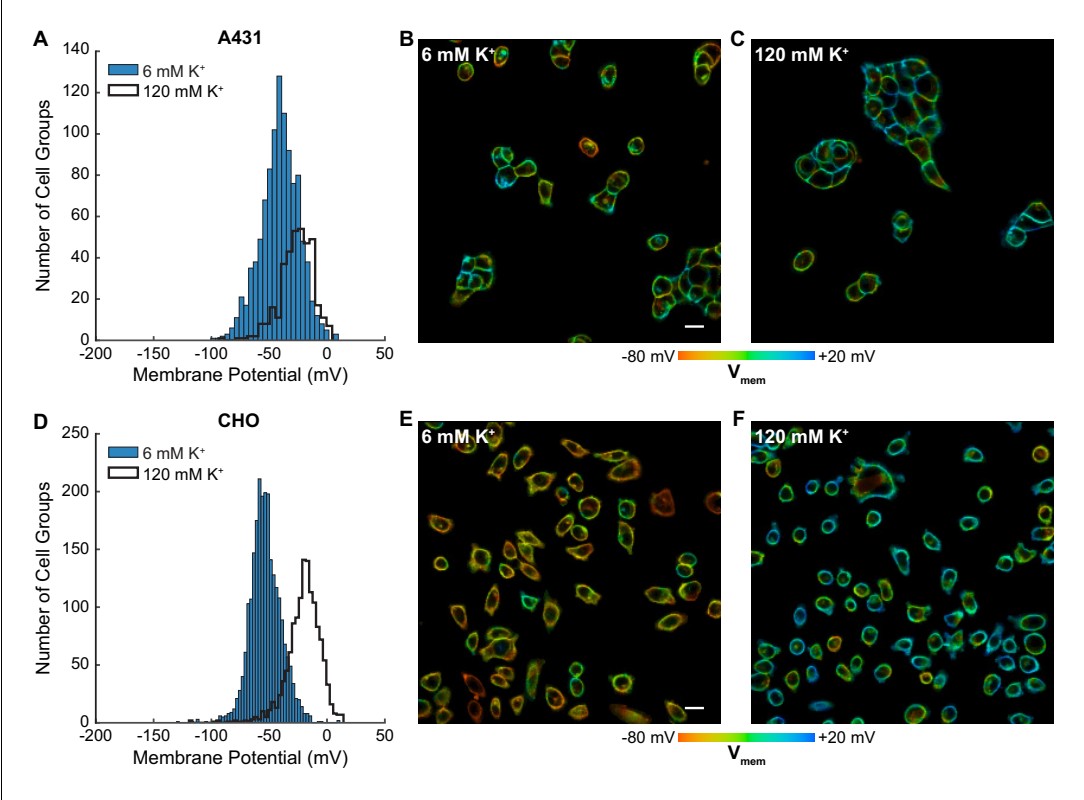

**Figure 3.** Rapid optical profiling of $V_{mem}$ at rest and in high extracellular $K^+$. Fluorescence lifetime images of cells incubated with 100 nM VF2.1.Cl were used to determine $V_{mem}$ from previously performed electrophysiological calibration (*Figure 2*). (A) Histograms of $V_{mem}$ values recorded in A431 cells incubated with 6 mM extracellular $K^+$ (commercial HBSS, n = 1056) or 120 mM $K^+$ (high $K^+$ HBSS, n = 368). (B) Representative lifetime image of A431 cells in 6 mM extracellular $K^+$. (C) Representative lifetime image of A431 cells in 120 mM extracellular $K^+$. (D) Histograms of $V_{mem}$ values observed in CHO cells under normal (n = 2410) and high $K^+$ (n = 1310) conditions. Representative lifetime image of CHO cells in (E) 6 mM and (F) 120 mM extracellular $K^+$. Histogram bin sizes were determined by the Freedman-Diaconis rule. Intensities in the lifetime-intensity overlay images are not scaled to each other. Scale bars, 20 μm.

DOI: https://doi.org/10.7554/eLife.44522.019

The following source data and figure supplements are available for figure 3:

**Source data 1.** $V_{mem}$ measurements made with VF-FLIM agree with previously reported values.
DOI: https://doi.org/10.7554/eLife.44522.022
**Figure supplement 1.** Optically recorded $V_{mem}$ distributions in HEK293T, MCF-7 and MDA-MB-231 cells.
DOI: https://doi.org/10.7554/eLife.44522.020
**Figure supplement 2.** Representative images of cultured cell resting membrane potential.
DOI: https://doi.org/10.7554/eLife.44522.021

We find that treatment of A431 cells with EGF results in a 15 mV hyperpolarization within 60–90 s in approximately 80% of cells (*Figure 4A–C*, *Figure 4—figure supplement 1*, *Figure 4—figure supplement 2*), followed by a slow return to baseline within 15 min (*Figure 4D–F*, *Figure 4—figure supplements 3* and 0 second acquisitions). The voltage response to EGF is dose-dependent, with an $EC_{50}$ of 90 ng/mL (14 nM) (*Figure 4—figure supplement 4*). Vehicle-treated cells show very little $\tau_{fl}$ change (*Figure 4A–F*). Identical experiments with voltage-insensitive VF2.0.Cl (*Figure 4G–H*, *Figure 4—figure supplement 1*, *Figure 4—figure supplement 3*, *Figure 4—figure supplement 5*) reveal little change in $\tau_{fl}$ upon EGF treatment, indicating the drop in $\tau_{fl}$ arises from membrane hyperpolarization. We observe the greatest hyperpolarization 1 to 3 min after treatment with EGF, which is abolished by inhibition of EGFR and ErbB2 tyrosine kinase activity with the covalent inhibitor canertinib (*Figure 4I–J*, *Figure 4—figure supplement 6*). Blockade of the EGFR kinase domain with gefitinib, a non-covalent inhibitor of EGFR, also results in a substantial decrease in the EGF-evoked hyperpolarization (*Figure 4I–J*, *Figure 4—figure supplement 6*). Together, these results indicate

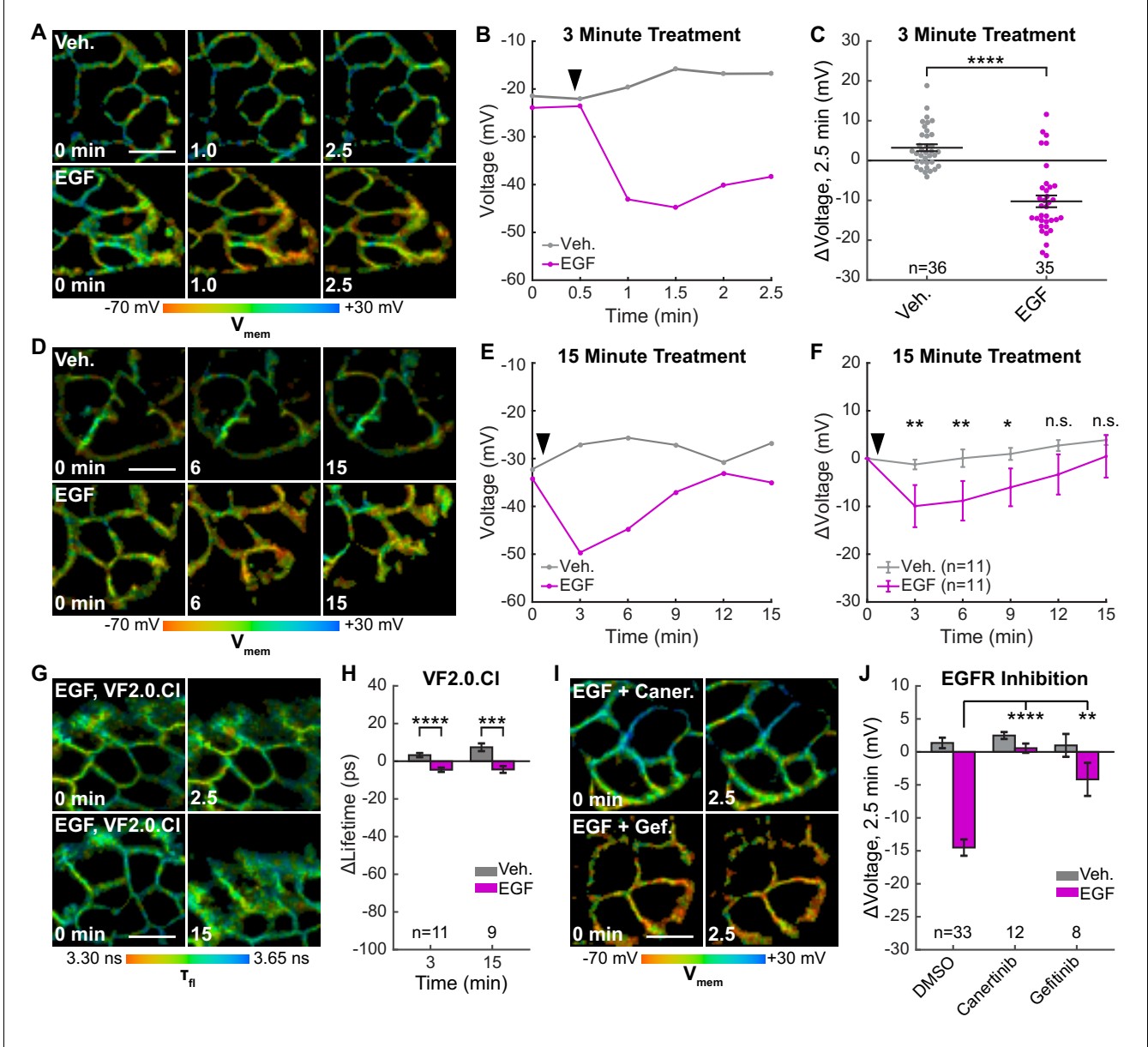

**Figure 4.** EGFR-mediated receptor tyrosine kinase activity produces a transient hyperpolarization in A431 cells. (A) Representative VF-FLIM time series of A431 cells treated with imaging buffer vehicle (top) or 500 ng/mL EGF (80 nM, bottom). (B) Quantification of images in (A), with Vehicle (Veh.)/EGF added at black arrow. (C) Aggregated responses for various trials of cells treated with vehicle or EGF. (D) Lifetime images of longer-term effects of vehicle (top) or EGF (bottom) treatment. (E) Quantification of images in (D). (F) Average response of cells over the longer time course. (G) Images of VF2.0.Cl (voltage insensitive) lifetime before and after EGF treatment. No $\tau_{fl}$ change is observed 2.5 (top) or 15 min (bottom) following EGF treatment. (H) Average VF2.0.Cl lifetime changes following EGF treatment. VF2.0.Cl graphs and images are scaled across the same lifetime range (350 ps) as VF2.1.Cl plots and images. The small drift observed would correspond to 2–4 mV of voltage change in VF2.1.Cl lifetime. (I) Lifetime images of A431 cells before and after EGF addition, with 500 nM canertinib (top) or 10 μM gefitinib (bottom). (J) Voltage changes 2.5 min after EGF addition in cells treated with DMSO (vehicle control) or an EGFR inhibitor. Scale bars are 20 μm. (C,F,H): Asterisks indicate significant differences between vehicle and EGF at that time point. (J): Asterisks reflect significant differences between EGF-induced voltage responses with DMSO vehicle or an EGFR inhibitor (n.s. p>0.05, *p<0.05, **p<0.01, ***p<0.001, ****p<0.0001, two-tailed, unpaired, unequal variances t-test).
DOI: https://doi.org/10.7554/eLife.44522.023

The following figure supplements are available for figure 4:

**Figure supplement 1.** Individual VF-FLIM recordings of A431 EGF response.
DOI: https://doi.org/10.7554/eLife.44522.024

**Figure supplement 2.** Membrane potential changes in A431 cells 2.5 min after EGF treatment.

*Figure 4 continued on next page*

*Figure 4 continued*

DOI: https://doi.org/10.7554/eLife.44522.025

**Figure supplement 3.** VF-FLIM reports A431 $V_{mem}$ changes over 15 min.

DOI: https://doi.org/10.7554/eLife.44522.026

**Figure supplement 4.** Dose-response relationship of A431 voltage response to EGF.

DOI: https://doi.org/10.7554/eLife.44522.027

**Figure supplement 5.** Effect sizes of VF2.1.Cl and VF2.0.Cl response to EGF treatment.

DOI: https://doi.org/10.7554/eLife.44522.029

**Figure supplement 6.** EGFR inhibitors abolish voltage response to EGF in A431 cells.

DOI: https://doi.org/10.7554/eLife.44522.028

that A431 cells exhibit an EGF-induced hyperpolarization, which depends on the kinase activity of EGFR and persists on the timescale of minutes.

Outward $K^+$ currents could mediate EGF-induced hyperpolarization. Consistent with this hypothesis, dissipation of the $K^+$ driving force by raising extracellular $[K^+]$ completely abolishes the typical hyperpolarizing response to EGF and instead results in a small depolarizing potential of approximately 3 mV (*Figure 5A*, *Figure 5—figure supplement 1B*). Blockade of voltage-gated $K^+$ channels ($K_v$) with 4-aminopyridine (4-AP) prior to EGF treatment enhances the hyperpolarizing response to EGF (*Figure 5A and B*, *Figure 5—figure supplement 1C*). In contrast, blockade of $Ca^{2+}$-activated $K^+$ channels ($K_{Ca}$) with charybdotoxin (CTX) results in a depolarizing potential of approximately 4 mV after exposure to EGF, similar to that observed with high extracellular $[K^+]$ (*Figure 5A and B*, *Figure 5—figure supplement 1D*). TRAM-34, a specific inhibitor of the intermediate-conductance $Ca^{2+}$ activated potassium channel $K_{Ca}3.1$ (*Wulff et al., 2000*), also abolishes EGF-induced hyperpolarization (*Figure 5A*, *Figure 5—figure supplement 1E*). CTX treatment has little effect on the resting membrane potential, while TRAM-34 or 4-AP depolarizes cells by approximately 5–10 mV (*Figure 5—figure supplement 2*).

To explore the effects of other components of the EGFR pathway on EGF-induced hyperpolarization, we perturbed intra- and extracellular $Ca^{2+}$ concentrations during EGF stimulation. Reduction of extracellular $Ca^{2+}$ concentration did not substantially alter the EGF response (*Figure 5A*, *Figure 5—figure supplement 1F*). However, sequestration of intracellular $Ca^{2+}$ with BAPTA-AM disrupts the hyperpolarization response. BAPTA-AM treated cells show a small, 4 mV depolarization in response to EGF treatment, similar to CTX-treated cells (*Figure 5A*, *Figure 5—figure supplement 1G*). Perturbation of $Ca^{2+}$ levels had little effect on the resting membrane potential (*Figure 5—figure supplement 2*). Introduction of wortmannin (1 µM) to block downstream kinase activity has no effect on the membrane potential response to EGF, while orthovanadate addition ($Na_3VO_4$, 100 µM) to block phosphatase activity results in a small increase in the hyperpolarizing response (*Figure 5A*, *Figure 5—figure supplement 1H–I*). These results support a model for EGF-EGFR mediated hyperpolarization in which RTK activity of EGFR causes release of internal $Ca^{2+}$ stores to in turn open $K_{Ca}$ channels and hyperpolarize the cell (*Figure 5C*).

## Discussion

We report the design and implementation of a new method for optically quantifying absolute membrane potential in living cells. VF-FLIM enjoys 100-fold improved throughput over patch clamp electrophysiology, as well as improved spatial resolution. The performance of VF-FLIM hinges on a balance between resolution in three dimensions: membrane potential, space, and time. We discuss the advantages and disadvantages of VF-FLIM in this light, as well as the new application space that is made accessible by VF-FLIM.

### Resolution of VF-FLIM: voltage, space, and time

The key advantage of VF-FLIM over previously reported optical approaches is its superior $V_{mem}$ resolution. Resolution can be interpreted as stability of the $\tau_{fl}$-$V_{mem}$ calibration over time and between cells. Any factors other than $V_{mem}$ that change $\tau_{fl}$ decrease resolution. VF-FLIM exhibits a 19-fold improvement in inter-cell $V_{mem}$ resolution over FLIM with the GEVI CAESR (*Brinks et al., 2015*) and a 8-fold improvement over di-8-ANEPPS excitation ratios (*Zhang et al., 1998*). Although all optical

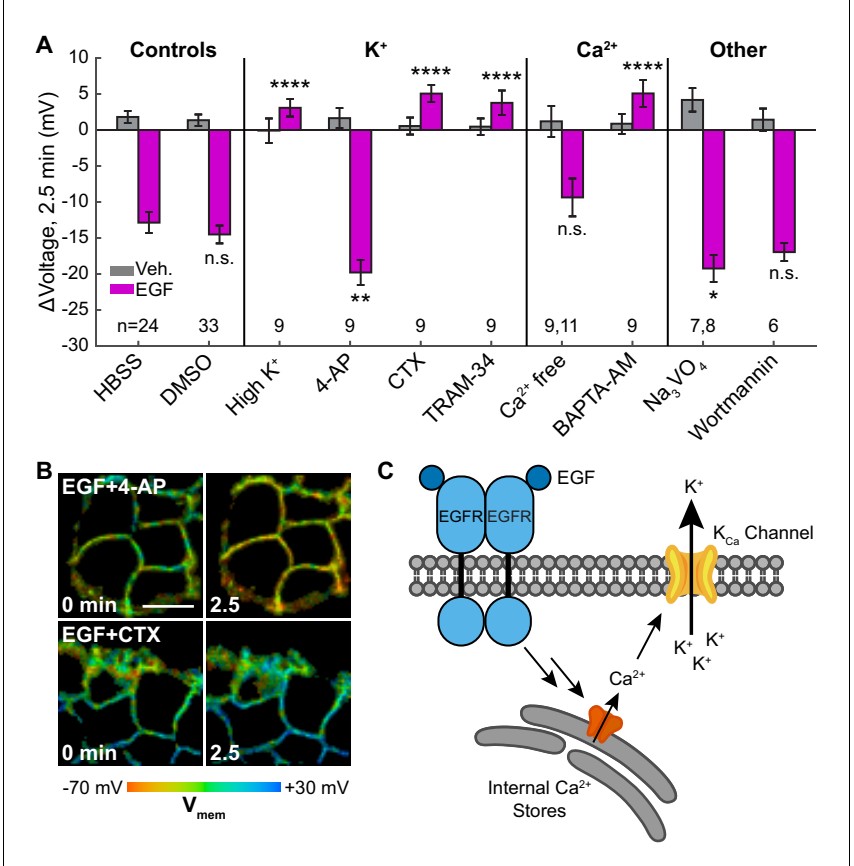

**Figure 5.** EGF-induced hyperpolarization is mediated by a $Ca^{2+}$ activated $K^+$ channel. (**A**) Comparison of the $V_{mem}$ change 2.5 min after EGF addition in cells incubated in unmodified imaging buffer (HBSS) or in modified solutions. (**B**) Lifetime images of A431 cells treated with 4-AP or CTX. (**C**) Model for membrane hyperpolarization following EGFR activation. Scale bar is 20 µm. Bars are mean ± SEM. Sample sizes listed are (Veh, EGF); where only one number is given, sample size was the same for both. Asterisks reflect significant differences in EGF-stimulated $V_{mem}$ change between the unmodified control (HBSS or DMSO) and modified solutions (n.s. $p > 0.05$, *$p < 0.05$, **$p < 0.01$, ***$p < 0.001$, ****$p < 0.0001$, two-tailed, unpaired, unequal variances t-test). DMSO: 0.1% DMSO, high $K^+$: 120 mM $K^+$, 4-AP: 5 mM 4-aminopyridine, CTX: 100 nM charybdotoxin, TRAM-34: 200 nM TRAM-34, $Ca^{2+}$ free: 0 mM $Ca^{2+}$ and $Mg^{2+}$, BAPTA-AM: 10 µM bisaminophenoxyethanetetraacetic acid acetoxymethyl ester, $Na_3VO_4$: 100 µM sodium orthovanadate, wortmannin: 1 µM wortmannin.

DOI: https://doi.org/10.7554/eLife.44522.030

The following figure supplements are available for figure 5:

**Figure supplement 1.** A431 voltage response to EGF with pharmacological intervention.
DOI: https://doi.org/10.7554/eLife.44522.031

**Figure supplement 2.** Effects of pharmacological and ionic perturbations on A431 resting membrane potential.
DOI: https://doi.org/10.7554/eLife.44522.032

---

strategies, including VF-FLIM, have worse $V_{mem}$ resolution than modern electrophysiology, the greater throughput, improved spatial resolution, and reduced invasiveness of optical strategies make them a powerful complement to electrode-based recordings.

The sources of variability that reduce resolution of optical $V_{mem}$ measurements are manifold, but two major contributors are membrane specificity of the stain and the complexity of the lipid environment. Nonspecific staining is fluorescence signal from anywhere other than the plasma membrane, such as contaminating intracellular staining from poorly trafficked (CAESR) or internalized (ANEPPS) sensor. In contrast, exogenously loaded VF2.1.Cl exhibits little fluorescence contribution from regions other than the plasma membrane. Secondly, membrane composition and dipole potential

can vary between cells and cell lines, changing the local environment of the fluorescent indicator (*Wang, 2012*; *Brügger, 2014*). Styryl dyes like di-8-ANEPPS can respond to changes in dipole potential (*Zhang et al., 1998*; *Gross et al., 1994*), and VF dyes may be similarly sensitive to dipole potential. Additionally, fluorescence lifetime depends on certain environmental factors (e.g. temperature, viscosity, ionic strength) (*Berezin and Achilefu, 2010*), which may introduce variability. These parameters are usually determined by the biological system under study, and re-calibration is important if they change dramatically in an experiment.

VF-FLIM, like all optical approaches, improves upon the spatial resolution of patch clamp electrophysiology. While VF-FLIM records the $V_{mem}$ of an optically defined region of interest (in this study a cell or cell group), electrophysiology records $V_{mem}$ at an individual cell or part of a cell where the electrode makes contact, which may or may not reflect the $V_{mem}$ of the entire cell or group. In this study, we interpret VF-FLIM at the whole cell level only, since that is the smallest unit in which the $V_{mem}$ can be reliably calibrated by whole cell patch clamp electrophysiology. Intriguingly, there are differences in lifetime within some cells in VF-FLIM images at the pixel to pixel level. In small, mostly spherical cells under voltage clamp, one would expect uniform membrane potential (*Armstrong and Gilly, 1992*), so these subcellular differences are most likely noise in the measurement. We speculate that most of this pixel-to-pixel noise comes from variability in fitting the biexponential lifetime model. Lifetime estimates at each pixel are calculated from 20 to 100-fold fewer photons than the lifetime value for the entire ROI. These lower photon counts at the single pixel level produce $V_{mem}$ estimates that are less precise than the $V_{mem}$ estimate for the entire ROI. Collection of more photons at each pixel could likely reduce this noise but would require longer acquisition times. We also cannot fully rule out an alternative explanation that the observed subcellular variability is the result of local differences in membrane composition (*Gross et al., 1994*).

$V_{mem}$ recordings in systems too large or too small for electrophysiological study could be an important application of VF-FLIM. Despite the improbability of $V_{mem}$ compartmentalization in individual HEK293T cells, other cells with complex morphology and processes may display real, subcellular $V_{mem}$ differences. In addition, delocalized $V_{mem}$ patterns across tissues could in theory be stable (*Cervera et al., 2016a*) and have been proposed to contribute to tissue development (*Levin, 2014*). One remaining challenge in expanding VF-FLIM to these areas is the requirement for an initial calibration with voltage clamp electrophysiology. Alternative ways to control $V_{mem}$, such as ionophores or optogenetic actuators (*Berndt et al., 2009*), may prove useful in these systems. When applying VF-FLIM to tissues, the cellular specificity of the VF stain becomes a consideration, as the VF2.1.Cl indicator used in this study labels all cell membranes efficiently. Looking ahead, recordings in tissue are an exciting area for future development of VF-FLIM, particularly in conjunction with cellular and sub-cellular strategies for targeting VF dyes (*Liu et al., 2017*; *Grenier et al., 2019*).

To obtain absolute $V_{mem}$ measurements with fluorescence lifetime, VF-FLIM sacrifices some of the temporal resolution of electrophysiology or intensity-based voltage imaging. VF-FLIM acquisition times are limited by the large numbers of photons needed per pixel in time-correlated single photon counting (see Materials and methods). As a result, VF-FLIM in its current implementation can track $V_{mem}$ events lasting longer than a few seconds. For 'resting' membrane potential or $V_{mem}$ dynamics associated with cell growth or differentiation, this temporal resolution is likely sufficient. Nevertheless, in the future, we envision allying VF-FLIM with recently developed, faster lifetime imaging technology to enable optical quantification of more rapid $V_{mem}$ responses (*Raspe et al., 2015*; *Gao et al., 2014*).

## Resting membrane potential distributions in cultured cells

Using the improved $V_{mem}$ resolution and throughput of VF-FLIM, we optically documented resting membrane potential distributions in cultured cells to characterize the membrane potential state(s) present. The presence and significance of distinct $V_{mem}$ states in cell populations is mostly uncharacterized due to the throughput limitations of patch-clamp electrophysiology, but some reports suggest that distinct $V_{mem}$ states arise during the various phases of the cell cycle (*Ouadid-Ahidouch et al., 2001*; *Wonderlin et al., 1995*). $V_{mem}$ histograms presented in this work appear more or less unimodal, showing no clear sign of cell cycle-related $V_{mem}$ states (*Figure 3A,D*; *Figure 3—figure supplement 1A,D,G*). We considered the possibility that VF-FLIM does not detect cell-cycle-related $V_{mem}$ states because we report average $V_{mem}$ across cell groups in cases where cells are in contact (*Figure 1—figure supplement 1*). This explanation is unlikely for two reasons.

First, $V_{mem}$ distributions for CHO cells appear unimodal, even though CHO cultures were mostly comprised of isolated cells under the conditions tested (*Figure 3D–F*). Second, theoretical work suggests that dramatically different $V_{mem}$ states in adjacent cells are unlikely, as electrical coupling often leads to equilibration of $V_{mem}$ across the cell group (*Cervera et al., 2016a*; *Cervera et al., 2016b*). Although we cannot rule out the possibility of poorly separated $V_{mem}$ populations (i.e. with a mean difference in voltage below our resolution limit), VF-FLIM both prompts and enables a re-examination of the notion that bi- or multimodal $V_{mem}$ distributions exist in cultured cells. Furthermore, VF-FLIM represents an exciting opportunity to experimentally visualize theorized $V_{mem}$ patterns in culture and in more complex tissues. Studies towards this end are ongoing in our laboratory.

## Epidermal growth factor induces $V_{mem}$ signaling in A431 cells

In the present study, we use VF-FLIM to provide the first cell-resolved, direct visualization of voltage changes induced by growth factor signaling. For long term $V_{mem}$ recordings during growth-related processes, an optical approach is more attractive than an electrode-based one. Electrophysiology becomes increasingly challenging as time scale lengthens, especially if cells migrate, and washout of the cytosol with pipette solution can change the very signals under study (*Horn and Korn, 1992*; *Malinow and Tsien, 1990*). Previous attempts to electrophysiologically record $V_{mem}$ in EGF-stimulated A431 cells were unsuccessful due to these technical challenges (*Pandiella et al., 1989*). Because whole cell voltage-clamp electrophysiology was intractable, the $V_{mem}$ response in EGF-stimulated A431 cells was addressed indirectly through model cell lines expressing EGFR exogenously (*Pandiella et al., 1989*), bulk measurements on trypsinized cells in suspension (*Magni et al., 1991*), or cell-attached single channel recordings (*Peppelenbosch et al., 1991*; *Lückhoff and Clapham, 1994*; *Mozhayeva et al., 1989*). By stably recording $V_{mem}$ during EGF stimulation, VF-FLIM enables direct study of $V_{mem}$ signaling in otherwise inaccessible pathways.

In conjunction with physiological manipulations and pharmacological perturbations, we explore the molecular mechanisms underlying EGF-induced hyperpolarization. We find that signaling along the EGF-EGFR axis results in a robust hyperpolarizing current carried by $K^+$ ions, passed by the $Ca^{2+}$-activated $K^+$ channel $K_{Ca}3.1$, and mediated by intracellular $Ca^{2+}$ (*Figure 5C*). We achieve a complete loss of the hyperpolarizing response to EGF by altering the $K^+$ driving force ('High $K^+$' *Figure 5A*, *Figure 5—figure supplement 1B*), blocking calcium-activated $K^+$ currents directly ('CTX' and 'TRAM-34', *Figure 5A*, *Figure 5—figure supplement 1D,E*), or intercepting cytosolic $Ca^{2+}$ ('BAPTA-AM', *Figure 5A*, *Figure 5—figure supplement 1G*). These results, combined with transcriptomic evidence that $K_{Ca}3.1$ is the major $K_{Ca}$ channel in A431 cells (*Thul et al., 2017*), indicate that $K_{Ca}3.1$ mediates the observed hyperpolarization. Interestingly, under some conditions where $K^+$-mediated hyperpolarization is blocked ('CTX,' 'high $K^+$', 'BAPTA-AM'), VF-FLIM reveals a small, secondary depolarizing current not visible during normal EGF stimulation. This current likely arises from initial $Ca^{2+}$ entry into the cell, as previously observed during EGF signaling (*Pandiella et al., 1987*; *Marquèze-Pouey et al., 2014*). Although we have obtained direct and conclusive evidence of EGF-induced hyperpolarization in A431 cells, the interactions between this voltage change and downstream targets of EGFR remain incompletely characterized. Enhancing EGF signaling by blockade of cellular tyrosine phosphatases with orthovanadate (*Reddy et al., 2016*) correspondingly increases EGF-mediated hyperpolarization ('$Na_3VO_4$' *Figure 5A*, *Figure 5—figure supplement 1H*), but inhibition of downstream kinase activity appears to have little effect on hyperpolarization ('wortmannin' *Figure 5A*, *Figure 5—figure supplement 1I*).

In the context of RTK signaling, $V_{mem}$ may serve to modulate the driving force for external $Ca^{2+}$ entry (*Huang and Jan, 2014*; *Yang and Brackenbury, 2013*) and thereby act as a regulator of this canonical signaling ion. Alternatively, $V_{mem}$ may play a more subtle biophysical role, such as potentiating lipid reorganization in the plasma membrane (*Zhou et al., 2015*). Small changes in $V_{mem}$ likely affect signaling pathways in ways that are currently completely unknown, but high throughput discovery of $V_{mem}$ targets remains challenging. Combination of electrophysiology with single cell transcriptomics has begun to uncover relationships between $V_{mem}$ and other cellular pathways in excitable cells (*Cadwell et al., 2016*); such approaches could be coupled to higher throughput VF-FLIM methods to explore pathways that interact with $V_{mem}$ in non-excitable contexts.

VF-FLIM represents a novel and general approach for interrogating the roles of membrane potential in fundamental cellular physiology. Future improvements to the voltage resolution could be made by use of more sensitive indicators, which may exhibit larger changes in fluorescence lifetime

(*Woodford et al., 2015*). VF-FLIM can be further expanded to include the entire color palette of PeT-based voltage indicators (*Huang et al., 2015*; *Deal et al., 2016*), allied with targeting methods to probe absolute membrane potential in heterogeneous cellular populations (*Liu et al., 2017*; *Grenier et al., 2019*), and coupled to high-speed imaging techniques for optical quantification of fast voltage events (*Raspe et al., 2015*; *Gao et al., 2014*).

# Materials and methods

**Key resources table**

| Reagent type (species) or resource | Designation | Source or reference | Identifiers | Additional information |
|---|---|---|---|---|
| Cell line (*Homo sapiens*, female) | A431 | UC Berkeley Cell Culture Facility | RRID:CVCL_0037 | Cell line maintained in E. Miller lab |
| Cell line (*Homo sapiens*, female) | HEK293T | UC Berkeley Cell Culture Facility | RRID:CVCL_0063 | Cell line maintained in E. Miller lab |
| Cell line (*Homo sapiens*, female) | MCF-7 | UC Berkeley Cell Culture Facility | RRID:CVCL_0031 | Cell line maintained in E. Miller lab |
| Cell line (*Homo sapiens*, female) | MDA-MB-231 | UC Berkeley Cell Culture Facility | RRID:CVCL_0062 | Cell line maintained in E. Miller lab |
| Cell line (*Cricetulus griseus*, female) | CHO | UC Berkeley Cell Culture Facility | RRID:CVCL_0214 | Cell line maintained in E. Miller lab |
| Recombinant DNA reagent | CAESR, FCK-QuasAR2-Citrine | Addgene, PMID:25118186 | Addgene:59172, RRID:Addgene_59172 | Developed by Adam Cohen, Harvard University |
| Peptide, recombinant protein | Recombinant human epidermal growth factor (EGF) | PeproTech | Cat#:AF100 15500UG | |
| Commercial assay or kit | Lipofectamine 3000 | Thermo Fisher Scientific | Cat#:L3000008 | |
| Commercial assay or kit | QIAprep spin miniprep kit | VWR International | Cat#:27106 | |
| Chemical compound, drug | Sodium orthovanadate | Sigma-Aldrich | CAS:13721-39-6, Cat#:S6508 | Activated before use (*Gordon, 1991*) |
| Chemical compound, drug | Canertinib | other | CAS:267243-28-7 | Gift from John Kuriyan, UC Berkeley |
| Chemical compound, drug | Gefitinib | Fisher Scientific | CAS:184475-35-2, Cat#:50-101-6270 | |

*Continued on next page*

*Continued*

| Reagent type (species) or resource | Designation | Source or reference | Identifiers | Additional information |
|---|---|---|---|---|
| Chemical compound, drug | 4-amino pyridine, 4-AP | Sigma-Aldrich | CAS:504-24-5, Cat#:A78403 | |
| Chemical compound, drug | Charybd otoxin, CTX | Sigma-Aldrich | CAS:95751-30-7, Cat#:C7802 | |
| Chemical compound, drug | TRAM-34 | Sigma-Aldrich | CAS:289905-88-0, Cat#:T6700 | |
| Chemical compound, drug | BAPTA-AM, bisamino-phenoxy-ethanetetra-acetic acid acetoxy methyl ester | Fisher Scientific | CAS:126150-97-8, Cat#:50-1 01-0334 | |
| Chemical compound, drug | wortmannin | Fisher Scientific | CAS:19545-26-7, Cat#: ICN19 569001 | |
| Software, algorithm | SPCM | Becker and Hickl | | |
| Other | Di-8-ANEPPS | Thermo Fisher Scientific | CAS:157134-53-7, Cat#:D3167 | |
| Other | VF2.1.Cl | Synthesized in-house (*Woodford et al., 2015*) | | |
| Other | VF2.0.Cl | Synthesized in-house (*Woodford et al., 2015*) | | |

VoltageFluor (VF) dyes VF2.1.Cl and VF2.0.Cl were synthesized in house according to previously described syntheses (*Woodford et al., 2015*). VFs were stored either as solids at room temperature or as 1000x DMSO stocks at −20°C. VF stock concentrations were normalized to the absorption of the dichlorofluorescein dye head via UV-Vis spectroscopy in Dulbecco's phosphate buffered saline (dPBS, Thermo Fisher Scientific, Waltham, MA) pH 9 with 0.1% sodium dodecyl sulfate (w/v, SDS). Di-8-ANEPPS was purchased from Thermo Fisher Scientific. Di-8-ANEPPS was prepared as a 2 mM (2000x) stock solution in DMSO and stored at −20°C. Di-8-ANEPPS concentrations were determined via UV-Vis spectroscopy in methanol ($\varepsilon$ at 498 nm: 41,000 $cm^{-1}$ $M^{-1}$ according to the manufacturer's certificate of analysis).

All salts and buffers were purchased from either Sigma-Aldrich (St. Louis, MO) or Fisher Scientific. TRAM-34, 4-aminopyridine, and charybdotoxin were purchased from Sigma-Aldrich. Gefitinib, wortmannin, sodium orthovanadate, and BAPTA-AM were purchased from Fisher Scientific. Canertinib was a gift from the Kuriyan laboratory at UC Berkeley. Gefitinib, wortmannin, canertinib, and TRAM-34 were made up as 1000x-10000x stock solutions in DMSO and stored at −20°C. Charybdotoxin was made up as a 1000x solution in water and stored at −80°C. 4-aminopyridine was made up as a 20x stock in imaging buffer (HBSS) and stored at 4°C. Recombinantly expressed epidermal growth factor was purchased from PeproTech (Rocky Hill, NJ) and aliquoted as a 1 mg/mL solution in water at −80°C.

Solid sodium orthovanadate was dissolved in water and activated before use (*Gordon, 1991*). Briefly, orthovanadate solutions were repeatedly boiled and adjusted to pH 10 until the solution was clear and colorless. 200 mM activated orthovanadate stocks were aliquoted and stored at −20°C.

Unless otherwise noted, all imaging experiments were performed in Hank's Balanced Salt Solution (HBSS; Gibco/Thermo Fisher Scientific). HBSS composition in mM: 137.9 NaCl, 5.3 KCl, 5.6

D-glucose, 4.2 NaHCO$_3$, 1.3 CaCl$_2$, 0.49 MgCl$_2$, 0.44 KH$_2$PO$_4$, 0.41 MgSO$_4$, 0.34 Na$_2$HPO$_4$. High K$^+$ HBSS was made in-house to 285 mOsmol and pH 7.3, containing (in mM): 120 KCl, 23.3 NaCl, 5.6 D-glucose, 4.2 NaHCO$_3$, 1.3 CaCl$_2$, 0.49 MgCl$_2$, 0.44 KH$_2$PO$_4$, 0.41 MgSO$_4$, 0.34 Na$_2$HPO$_4$. Nominally Ca$^{2+}$/Mg$^{2+}$ free HBSS (Gibco) contained, in mM: 137.9 NaCl, 5.3 KCl, 5.6 D-glucose, 4.2 NaHCO$_3$, 0.44 KH$_2$PO$_4$, 0.34 Na$_2$HPO$_4$.

## Methods

### Cell culture

All cell lines were obtained from the UC Berkeley Cell Culture Facility and discarded after twenty-five passages. A431, HEK293T, MCF-7, and MDA-MB-231 cells were authenticated by short tandem repeat (STR) profiling. All cells were routinely tested for mycoplasma contamination. Cells were maintained in Dulbecco's Modified Eagle Medium (DMEM) with 4.5 g/L D-glucose supplemented with 10% FBS (Seradigm (VWR); Radnor, PA) and 2 mM GlutaMAX (Gibco) in a 5% CO$_2$ incubator at 37°C. Media for MCF-7 cells was supplemented with 1 mM sodium pyruvate (Life Technologies/Thermo Fisher Scientific) and 1x non-essential amino acids (Thermo Fisher Scientific). Media for CHO.K1 (referred to as CHO throughout the text) cells was supplemented with 1x non-essential amino acids. HEK293T and MDA-MB-231 were dissociated with 0.05% Trypsin-EDTA with phenol red (Thermo Fisher Scientific) at 37°C, whereas A431, CHO, and MCF-7 cells were dissociated with 0.25% Trypsin-EDTA with phenol red at 37°C. To avoid potential toxicity of residual trypsin, all cells except for HEK293T were spun down at 250xg or 500xg for 5 min and re-suspended in fresh complete media during passaging.

For use in imaging experiments, cells were plated onto 25 mm diameter poly-D-lysine coated #1.5 glass coverslips (Electron Microscopy Sciences) in six well tissue culture plates (Corning; Corning, NY). To maximize cell attachment, coverslips were treated before use with 1–2 M HCl for 2–5 hr and washed overnight three times with 100% ethanol and three times with deionized water. Coverslips were sterilized by heating to 150°C for 2–3 hr. Before use, coverslips were incubated with poly-D-lysine (Sigma-Aldrich, made as a 0.1 mg/mL solution in phosphate-buffered saline with 10 mM Na$_3$BO$_4$) for 1–10 hr at 37°C and then washed twice with water and twice with Dulbecco's phosphate buffered saline (dPBS, Gibco).

A431, CHO, HEK293T, and MCF-7 were seeded onto glass coverslips 16–24 hr before microscopy experiments. MDA-MB-231 cells were seeded 48 hr before use because it facilitated formation of gigaseals during whole-cell voltage clamp electrophysiology. Cell densities used for optical resting membrane potential recordings (in 10$^3$ cells per cm$^2$) were: A431 42; CHO 42; HEK293T 42; MCF-7 63; MDA-MB-231 42. To ensure the presence of single cells for whole-cell voltage clamp electrophysiology, fast-growing cells were plated more sparsely (approximately 20% confluence) for electrophysiology experiments. Cell densities used for electrophysiology (in 10$^3$ cells per cm$^2$) were: A431 36–52; CHO 21; HEK293T 21; MCF-7 63; MDA-MB-231 42. To reduce their rapid growth rate, HEK293T cells were seeded onto glass coverslips in reduced glucose (1 g/L) DMEM with 10% FBS, 2 mM GlutaMAX, and 1 mM sodium pyruvate for electrophysiology experiments.

### Cellular loading of VoltageFluor dyes

Cells were loaded with 1x VoltageFluor in HBSS for 20 min in a 37°C incubator with 5% CO$_2$. For most experiments, 100 nM VoltageFluor was used. Serum-starved A431 cells were loaded with 50 nM VoltageFluor. After VF loading, cells were washed once with HBSS and then placed in fresh HBSS for imaging. All imaging experiments were conducted at room temperature under ambient atmosphere. Cells were used immediately after loading the VF dye, and no cells were kept for longer than an hour at room temperature.

### Whole-cell patch-clamp electrophysiology

Pipettes were pulled from borosilicate glass with filament (Sutter Instruments, Novato, CA) with resistances ranging from 4 to 7 MΩ with a P97 pipette puller (Sutter Instruments). Internal solution composition, in mM (pH 7.25, 285 mOsmol/L): 125 potassium gluconate, 10 KCl, 5 NaCl, 1 EGTA, 10 HEPES, 2 ATP sodium salt, 0.3 GTP sodium salt. EGTA (tetraacid form) was prepared as a stock solution in either 1 M KOH or 10 M NaOH before addition to the internal solution. Pipettes were positioned with an MP-225 micromanipulator (Sutter Instruments). A liquid junction potential of −14

mV was determined by the Liquid Junction Potential Calculator in the pClamp software package (*Barry, 1994*) (Molecular Devices, San Jose, CA), and all voltage step protocols were corrected for this offset.

For VF-FLIM and CAESR, electrophysiology recordings for VF-FLIM and CAESR were made with an Axopatch 200B amplifier and digitized with a Digidata 1440A (Molecular Devices). The software package used was pClamp 10.3. Signals were filtered with a 5 kHz low-pass Bessel filter. Correction for pipette capacitance was performed in the cell attached configuration. Voltage-lifetime calibrations were performed in V-clamp mode, with the cell held at the potential of interest for 15 or 30 s while lifetime was recorded. Potentials were applied in random order, and membrane test was conducted between each step to verify the quality of the patch. For single cell patching, recordings were only included if they maintained a 30:1 ratio of membrane resistance ($R_m$) to access resistance ($R_a$) and an $R_a$ value below 30 MΩ throughout the recording. Due to the reduced health of HEK293T cells transfected with CAESR, recordings were used as long as they maintained a 10:1 $R_m$:$R_a$ ratio, although most recordings were better than 30:1 $R_m$:$R_a$. Only recordings stable for at least four voltage steps (roughly 2 min) were included in the dataset.

For di-8-ANEPPS, electrophysiology recordings were made in the same manner as the above, with the following minor differences. Signals were digitized with a Digidata 1550B; the pClamp 10.6 software package was used (Molecular Devices). Potentials were applied in the order 0 mV, −80 mV, +40 mV, −40 mV, +80 mV for ten seconds at each step. Patch parameters were tested at the beginning and end of the patch program, rather than between each step. Only patches that retained a 30:1 ratio of Rm to Ra and access resistance below 30 MΩ throughout the recording were included in the dataset.

For electrophysiology involving small groups of cells (*Figure 2—figure supplement 3*), complete voltage clamp across the entire cell group was not possible. Recordings were used as long as $R_a$ remained below 30 MΩ for at least three voltage steps. Most recordings also retained $R_m$:$R_a$ ratios greater than 20:1.

## Epidermal growth factor treatment

A431 cells were serum starved prior to epidermal growth factor studies. Two days before the experiment, cells were trypsinized and suspended in complete media with 10% FBS. Cells were then spun down for 5 min at 500xg and re-suspended in reduced serum DMEM (2% FBS, 2 mM GlutaMAX, 4.5 g/L glucose). Cells were seeded onto 25 mm coverslips in six well plates at a density of $84 \times 10^3$ cells per cm$^2$. 4–5.5 hr before the experiment, the media was exchanged for serum-free DMEM (0% FBS, 2 mM GlutaMAX, 4.5 g/L glucose).

After 4–5.5 hr in serum-free media, cells were loaded with 50 nM VF dye as described above. In pharmacology experiments, the drug or vehicle was also added to the VF dye loading solution. All subsequent wash and imaging solutions also contained the drug or vehicle. For changes to buffer ionic composition, VoltageFluor dyes were loaded in unmodified HBSS to avoid toxicity from prolonged incubation with high K$^+$ or without Ca$^{2+}$. Immediately prior to use, cells were washed in the modified HBSS (120 mM K$^+$ or 0 mM Ca$^{2+}$) and recordings were made in the modified HBSS.

For analysis of short-term responses to EGF (3 min time series), VF lifetime was recorded in 6 sequential 30 s exposures. Immediately after the conclusion of the first frame (30–35 s into the recording), EGF or vehicle (imaging buffer only) was added to the indicated final concentration from a 2x solution in HBSS imaging buffer. For analysis of long-term responses to EGF (15 min time series), EGF addition occurred in the same way, but a gap of 150 s (without laser illumination) was allotted between each 30 s lifetime recording. Times given throughout the text correspond to the start of an exposure. Voltage changes at 2.5 min were calculated from the difference between an initial image (taken before imaging buffer vehicle or EGF addition) and a final image (a 30 s exposure starting 2.5 min into the time series).

## Transfection and imaging of CAESR in HEK293T

The CAESR plasmid was obtained as an agar stab (FCK-Quasar2-Citrine, Addgene #59172), cultured overnight in LB with 100 µg/mL ampicillin, and isolated via a spin miniprep kit (Qiagen). HEK293T cells were plated at a density of 42,000 cells per cm$^2$ directly onto a six well tissue culture plate and incubated at 37°C in a humidified incubator for 24 hr prior to transfection. Transfections were

performed with Lipofectamine 3000 according to the manufacturer's protocol (Thermo Fisher Scientific). Cells were allowed to grow an additional 24 hr after transfection before they were plated onto glass coverslips for microscopy experiments (as described above for electrophysiology of untransfected HEK293T cells).

## Determination of $EC_{50}$ for EGF in A431 cells

Average voltage changes 2.5 min after addition of EGF to serum deprived A431 cells were determined at different EGF concentrations, and these means were fit to a four parameter logistic function in MATLAB (MathWorks, Natick, MA).

## Goldman-Hodgkin-Katz estimation of $V_{mem}$ ranges in different imaging buffers

If intracellular and extracellular concentrations, as well as relative permeabilities, of all ionic species are known, the Goldman-Hodgkin-Katz (GHK) equation (Equation 1) can be used to calculate the resting membrane potential of a cell (Hodgkin and Katz, 1949). In practice, the intracellular ion concentrations $[X]_{in}$ and relative permeabilities $P_x$ are difficult to determine, so the GHK equation is not a substitute for direct measurement of $V_{mem}$. To obtain a range of reasonable $V_{mem}$ values in systems where these concentrations and relative permeabilities are not known, we calculated possible $V_{mem}$ using the 'standard' parameters derived from Hodgkin and Katz (1949), as well as a value above and a value below each 'standard' point. The values evaluated were the following: $P_K$ 1; $P_{Na}$ 0.01, 0.05, 0.2; $P_{Cl}$ 0.2, 0.45, 0.9; $[K^+]_{in}$ 90, 150, 200 mM; $[Na^+]_{in}$ 5, 15, 50 mM; $[Cl^-]_{in}$ 2, 10, 35 mM. Extracellular ion concentrations $[X]_{out}$ were known (see Materials and methods). In Equation 1, R is the universal gas constant, T is the temperature (293 K for this experiment), and F is Faraday's constant.

$$V_{mem} = \frac{RT}{F} ln \frac{P_K[K^+]_{out} + P_{Na}[Na^+]_{out} + P_{Cl}[Cl^-]_{in}}{P_K[K^+]_{in} + P_{Na}[Na^+]_{in} + P_{Cl}[Cl^-]_{out}} \qquad (1)$$

## Fluorescence lifetime data acquisition

Fluorescence lifetime imaging was conducted on a LSM 510 inverted scanning confocal microscope (Carl Zeiss AG, Oberkochen, Germany) equipped with an SPC-150 or SPC-150N single photon counting card (Becker and Hickl GmbH, Berlin, Germany) (Scheme 1). 80 MHz pulsed excitation was supplied by a Ti:Sapphire laser (MaiTai HP; SpectraPhysics, Santa Clara, CA) tuned to 958 nm and frequency-doubled to 479 nm. The laser was cooled by a recirculating water chiller (Neslab KMC100). Excitation light was directed into the microscope with a series of silver mirrors (Thorlabs, Newton, NJ or Newport Corporation, Irvine, CA).

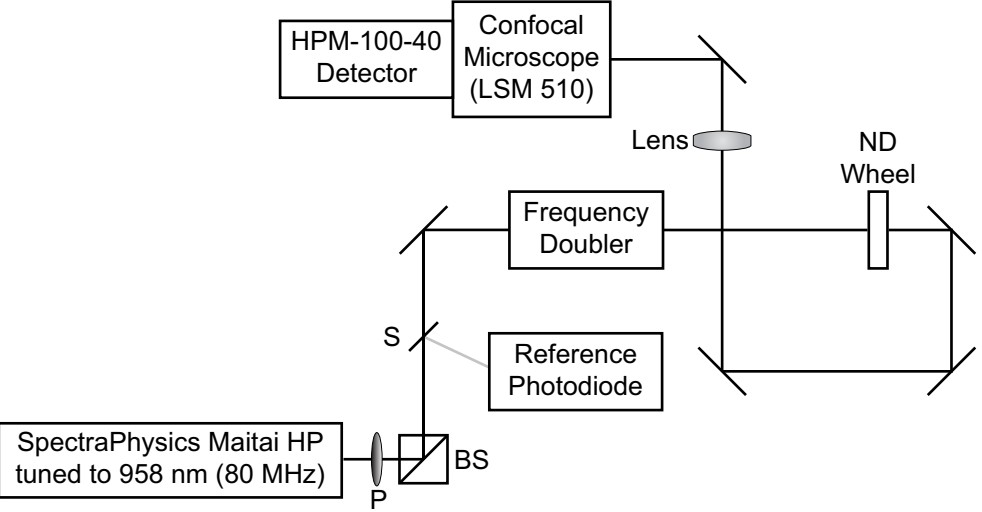

**Scheme 1.** Optical diagram for time correlated single photon counting microscope. Excitation light was supplied by a Ti:Sapphire laser tuned to 958 nm. A small amount of light was redirected by a beam sampler (S) to a reference photodiode. The remaining light was passed through a frequency doubler to obtain 479 nm excitation

light, which entered the LSM510 confocal microscope. A polarizer (P) followed by a polarizing beamsplitter (BS), as well as a neutral density (ND) wheel, allowed control of the amount of light passed to the sample.
DOI: https://doi.org/10.7554/eLife.44522.033

Excitation light power at the sample was controlled with a neutral density (ND) wheel and a polarizer (P) followed by a polarizing beamsplitter (BS). Light was titrated such that VoltageFluor lifetime did not drift during the experiment, no phototoxicity was visible, and photon pile-up was not visible on the detector. For recordings at high VoltageFluor concentrations (*Figure 1—figure supplement 2*, *Figure 2—figure supplement 4*), reduced power was used to avoid saturating the detector. For optical voltage determinations using 50 or 100 nM VoltageFluor, typical average power at the sample was 5 µW.

Fluorescence emission was collected through a 40x oil immersion objective (Zeiss) coated with immersion oil (Immersol 518F, Zeiss). Emitted photons were detected with a hybrid detector, HPM-100–40 (Becker and Hickl), based on a Hamamatsu R10467 GaAsP hybrid photomultiplier tube. Detector dark counts were kept below 1000 per second during acquisition. Emission light was collected through a 550/49 bandpass filter (Semrock, Rochester, NY) after passing through a 488 LP dichroic mirror (Zeiss). The reference photons for determination of photon arrival times were detected with a PHD-400-N high speed photodiode (Becker and Hickl). Data were acquired with 256 time bins in the analog-to-digital-converter and either $64 \times 64$ or $256 \times 256$ pixels of spatial resolution (see discussion of pixel size below).

Routine evaluation of the proper functioning of the lifetime recording setup was performed by measurement of three standards (*Figure 1—source data 2*): 2 µM fluorescein in 0.1 N NaOH, 1 mg/mL erythrosin B in water (pH 7), and the instrument response function (IRF). The IRF was determined from a solution of 500 µM fluorescein and 12.2 M sodium iodide in 0.1 N NaOH. Because of the high concentration of iodide quencher, the IRF solution has a lifetime shorter than the detector response time, allowing approximation of the instrument response function under identical excitation and emission conditions as data acquisition (*Liu et al., 2014*).

## IRF deconvolution

Signal from photons detected in a TCSPC apparatus are convolved with the instrument response (IRF). IRFs can be approximated by the SPCImage fitting software, but consistency of lifetime fits on VF-FLIM datasets was improved by using a measured IRF. Measured IRFs were incorporated by the iterative reconvolution method using SPCImage analysis software (*Becker, 2012*).

## VoltageFluor lifetime fitting model

All VoltageFluor lifetime data were fit using SPCImage (Becker and Hickl), which solves the nonlinear least squares problem using the Levenberg-Marquadt algorithm. VF2.1.Cl lifetime data were fit to a sum of two exponential decay components (*Equation 2*). Attempts to fit the VF2.1.Cl data with a single exponential decay (*Equation 3*) were unsatisfactory.

$$F(t) = a_1 e^{\frac{-t}{\tau_1}} + a_2 e^{\frac{-t}{\tau_2}} \tag{2}$$

The fluorescence lifetime of VF2.0.Cl was adequately described by a single exponential decay for almost all data (*Equation 3*). A second exponential component was necessary to fit data at VF2.0.Cl concentrations above 500 nM, likely attributable to the concentration-dependent decrease in lifetime that was observed high VF concentrations.

$$F(t) = a e^{\frac{-t}{\tau}} \tag{3}$$

For all data fit with the two component model, the weighted average of the two lifetimes, $\tau_m$ (*Equation 4*), was used in subsequent analysis.

$$\tau_m = \frac{a_1 \tau_1 + a_2 \tau_2}{a_1 + a_2} \tag{4}$$

All lifetime images are represented as an overlay of photon count (pixel intensity) and weighted average lifetime (pixel color) throughout the text ($\tau_m$ + PC, *Figure 1—figure supplement 1*). Pixels

with insufficient signal to fit a fluorescence decay are shown in black. The photon counts, as well as the lifetimes, in image sequences on the same set of cells are scaled across the same range.

## Additional fit parameters for VoltageFluor lifetimes

Pixels with photon counts below 300 (VF2.1.Cl) or 150 (VF2.0.Cl) photons at the peak of the decay (time bin with the most signal) were omitted from analysis to ensure reproducible fits. Because the lifetime of VFs does not fully decay to baseline in a single 12.5 ns laser cycle, the incomplete multiexponentials fitting option was used, allowing the model to attribute some signal early in the decay to the previous laser cycle. Out of 256 time bins from the analog-to-digital converter (ADC), only data from time bins 23 to 240 were used in the final fit. The offset parameter (detector dark counts per ADC time bin per pixel) was set to zero. The number of iterations for the fit in SPCImage was increased to 20 to obtain converged fits. Shift between the IRF and the decay trace was fixed to 0.5 (in units of ADC time bins), which consistently gave lifetimes of standards erythrosin B (1 mg/mL in $H_2O$) (*Boens et al., 2007*) and fluorescein (2 µM in 0.1 N NaOH, $H_2O$) (*Magde et al., 1999*) closest to reported values (*Figure 1—source data 2*).

## Acquisition time and effective pixel size in lifetime data

To obtain sufficient photons but keep excitation light power minimal, binning between neighboring pixels was employed during fitting. This procedure effectively takes the lifetime as a spatial moving average across the image by including adjacent pixels in the decay for a given pixel. To obtain larger photon counts, the confocal pinhole was set between 2.5 and 3.5 airy units, which corresponds to optical section thickness of approximately 2.5 µm.

| Data type | Acquired pixel width (µm) | Binned pixel width (µm) | Acquisition time (s) | Img size (pixels) |
|---|---|---|---|---|
| Concentration Curve (*Figure 1—figure supplement 2*, *Figure 2—figure supplement 4*) | 0.44 | 3.08 | 75–90 | 256 × 256 |
| $V_{mem}$ Distributions (*Figure 3*) | 1.24 | 8.68 | 90–120 | 256 × 256 |
| Electrophysiology Recording | 1.00 | 3.01 | 15–30 | 64 × 64 |
| EGF Time Series | 0.88 | 2.64 | 30 | 64 × 64 |

All tabulated values are for an individual frame, although multiple sequential frames were recorded in both the electrophysiology and EGF experiments. For each recording type, the width of each pixel at acquisition is reported, as well as the width of the area included in the binned lifetime signal during fitting. All pixels are square. The acquisition time reflects the total time to collect the image, not the total time exposing each pixel. All FLIM images have 256 time bins in the ns regime, so a 256 × 256 spatial image size represents a 256 × 256 × 256 total dataset. Img = image.

## Determination of regions of interest

Images were divided into cell groups, with each cell group as a single region of interest (ROI). ROIs were determined from photon count images, either manually from the cell morphology in FIJI (*Schindelin et al., 2012*) or automatically by sharpening and then thresholding the signal intensity with custom MATLAB code (*Source code 2*). Regions of images that were partially out of the optical section or contained punctate debris were omitted. Sample ROIs are shown in *Figure 1—figure supplement 1*.

For cells that adjoin other cells, attribution of a membrane region to one cell versus the other is not possible. As such, we chose to interpret each cell group as an independent sample ('n') instead of extracting $V_{mem}$ values for individual cells. Adjacent cells in a group are electrically coupled to varying degrees, and their resting membrane potentials are therefore not independent (*Meşe et al., 2007*). While this approach did not fully utilize the spatial resolution of VF-FLIM, it prevented overestimation of biological sample size for the effect in question.

## Conversion of lifetime to transmembrane potential

The mean $\tau_m$ across all pixels in an ROI was used as the lifetime for that ROI. Lifetime values were mapped to transmembrane potential via the lifetime-$V_{mem}$ standard curves determined with whole-cell voltage-clamp electrophysiology. For electrophysiology measurements, the relationship between the weighted average lifetime (*Equation 4*) and membrane potential for each patched cell was determined by linear regression, yielding a sensitivity (*m*, ps/mV) and a 0 mV lifetime (*b*, ps) for each cell (*Equation 5*). The average sensitivity and 0 mV point across all cells of a given type were used to convert subsequent lifetime measurements ($\tau$) to $V_{mem}$ (*Figure 2—source data 1*, *Equation 6*). For quantifying changes in voltage ($\Delta V_{mem}$) from changes in lifetime ($\Delta \tau$), only the average sensitivity is necessary (*Equation 7*).

$$\tau = m * V_{mem} + b \tag{5}$$

$$V_{mem} = \frac{(\tau - b)}{m} \tag{6}$$

$$\Delta V_{mem} = \frac{(\Delta \tau)}{m} \tag{7}$$

Where standard error of the mean of a voltage determination ($\delta V_{mem}$) is given, error was propagated to include the standard errors of the slope ($\delta m$) and y-intercept ($\delta b$) of the voltage calibration, as well as the standard error of the lifetime measurements ($\delta \tau$) in the condition of interest (*Equation 8*). For error in a voltage change ($\delta \Delta V_{mem}$), only error in the calibration slope was included in the propagated error (*Equation 9*). Where standard deviation of VF-FLIM derived $V_{mem}$ values is shown, a similar error propagation procedure was applied, using the standard deviation of the average sensitivity and 0 mV lifetime for that cell line.

$$\delta V_{mem} = |V_{mem}| \sqrt{\left(\frac{\sqrt{\delta \tau^2 + \delta b^2}}{\tau - b}\right)^2 + \left(\frac{\delta m}{m}\right)^2} \tag{8}$$

$$\delta \Delta V_{mem} = |\Delta V_{mem}| \sqrt{\left(\frac{\delta \Delta \tau}{\Delta \tau}\right)^2 + \left(\frac{\delta m}{m}\right)^2} \tag{9}$$

## Resolution of VF-FLIM voltage determination

The intrinsic nature of fluorescence lifetime introduces a point of reference into the voltage measurement, from which a single lifetime image can be interpreted as resting membrane potential. Differences in this reference point (reported here as the 0 mV lifetime) over time and across cells provides an estimate of the voltage-independent noise in VF-FLIM. We report resolution as the root-mean-square deviation (RMSD) between the optically calculated voltage ($V_{FLIM}$) and the voltage set by whole-cell voltage clamp ($V_{ephys}$), which is analogous to the resolution calculations described previously by Cohen and co-workers (*Hou et al., 2014*). The RMSD of n measurements (*Equation 10*) can be determined from the variance $\sigma$ (*Cone and Cone, 1976*) (*Equation 11*) and the bias (*Equation 12*) of the estimator (in this case, VF-FLIM) relative to the 'true' value (in this case, electrophysiology). These calculations are described graphically in *Scheme 2* below.

$$RMSD = \sqrt{\sigma^2 + Bias^2} \tag{10}$$

$$\sigma^2 = \frac{1}{n} \sum_{i=1}^{n} \left(V_{FLIM,i} - V_{ephys,i}\right)^2 \tag{11}$$

$$Bias = \frac{1}{n} \sum_{i=1}^{n} V_{FLIM,i} - \frac{1}{n} \sum_{i=1}^{n} V_{ephys,i} \tag{12}$$

The voltage-independent variations in lifetime are much larger between cells than within a cell.

Therefore, the error in measuring absolute voltage *changes* on a given cell ('intra-cell' comparisons) is lower than the error in determining the absolute $V_{mem}$ of that cell ('inter-cell' comparisons, since the calibration used is from another cell). We can therefore determine an 'intra-cell' RMSD and an 'inter-cell' RMSD to reflect the voltage resolution of these two types of measurements. To calculate 'intra-cell' error, we look at the RMSD between $V_{ephys}$ and $V_{FLIM}$ using the $\tau_{fl}$-$V_{mem}$ relationship *for that specific cell*. Phrased another way, we are looking at the amount of error that would be expected in estimating $V_{mem}$ of a cell if its exact $\tau_{fl}$-$V_{mem}$ relationship were known. This 'intra cell' RMSD estimates the error expected in quantifying changes in $V_{mem}$ on a given cell. We calculate an intra cell error for each cellular recording, so intra cell errors are reported throughout the text as a mean ± SEM of the intra cell errors for all individual cells of a given type. The average intra cell error was at or below 5 mV for all cell lines tested (*Figure 2—source data 1*).

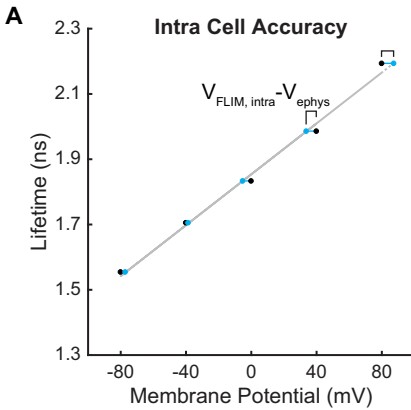

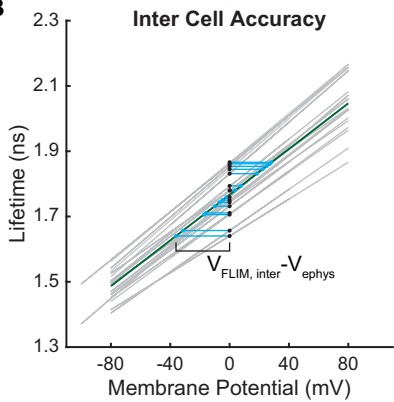

Black point: Measured τ at a set $V_{ephys}$
Blue point: $V_{FLIM, intra}$, the optical guess at $V_{mem}$ from the measured τ and the individual cell's line of best fit

*For each cell*, determine RMSD between $V_{FLIM}$ and $V_{ephys}$ at each potential

Average intra cell RMSD across all cells from a given line (report as mean ± SEM).

Black point: y-intercept (0 mV lifetime) from an individual cell's line of best fit
Blue point: $V_{FLIM, inter}$, the optical guess at $V_{mem}$ from those y-intercepts (*ideally* would be 0 mV) based on the average calibration for a cell type

*For all cells of a given type together*, determine RMSD between $V_{FLIM}$ and $V_{ephys}$ at 0 mV

**Scheme 2.** Intra and inter-cell $V_{mem}$ resolution calculations. Data are taken directly from *Figure 1H,I* as an example. (**A**) Intra cell values are the RMSD between the voltage equivalent of the measured lifetime ($V_{FLIM}$) and voltage set by electrophysiology ($V_{ephys}$). $V_{FLIM}$ values are calculated using that particular cell's line of best fit, so one value is obtained per cell. Here, we present intra cell error as the mean ± SEM of all cells from a given cell line. (**B**) Inter cell errors are the RMSD between the voltage-equivalent of the 0 mV lifetime for all cells tested from a cell line ($V_{FLIM}$, determined with the average slope and y-intercept for that cell line) and the ground truth value of 0 mV. Inter-cell accuracy is calculated from all of the calibration data for a cell line, so there is one value per cell line. Black points are experimental y-intercepts and blue points are the $V_{FLIM}$ optical voltage determinations from those lifetimes. Gray lines are lines of best fit for individual cells. Green line in (B) represents the average $\tau_{fl}$-$V_{mem}$ relationship for a cell line.
DOI: https://doi.org/10.7554/eLife.44522.012

The error in the absolute membrane potential determination ('inter-cell') is calculated here as the RMSD between the y-intercept (0 mV lifetime) of all of the individual cells' lifetime-voltage relationships and the 0 mV value for the averaged calibration *for all cells of a given type*. This metric quantifies how well the lifetime-$V_{mem}$ relationship for a given cell line represents an individual cell's lifetime-$V_{mem}$ relationship. This 'inter cell' RMSD ranged from 10 to 23 mV for the tested cell lines (*Figure 2—source data 1*). Much smaller errors for a population value of $V_{mem}$ can be obtained by averaging $V_{mem}$ recordings from multiple cells.

This method of calculating error assumes that the electrophysiology measurement is perfectly accurate and precise. Realistically, it is likely that some of the variation seen is due to the quality of the voltage clamp. As a result, these RMSD values provide a conservative upper bound for the voltage errors in VF-FLIM.

## Analysis of CAESR lifetimes

For sample images of CAESR in HEK293T (*Figure 1—figure supplement 4*), fluorescence decays were fit using SPCImage to a biexponential decay model as described for VF2.1.Cl above, using a peak photon threshold of 150 and a bin of 2 (binned pixel width of 5 µm). To better match the studies by Cohen and co-workers (*Brinks et al., 2015*), which isolated the membrane fluorescence from cytosolic fluorescence by directing the laser path, the lifetime-voltage relationships were not determined with these square-binned images. Instead, membranes were manually identified, and the fluorescence decays from all membrane pixels were summed together before fitting once per cell. (This is in contrast to the processing of VoltageFluor data, where the superior signal to noise and localization enables fitting and analysis of the lifetime on a pixel by pixel basis). This 'one fit per membrane' analysis of CAESR was performed in custom MATLAB code implementing a Nelder-Mead algorithm (*Source code 1*, adapted from *Enderlein and Erdmann, 1997*). CAESR data were fit to a biexponential model with the offset fixed to 0 and the color shift as a free parameter.

## Di-8-ANEPPS ratio-based imaging

In preparation for imaging, HEK293T cells were plated as described above for electrophysiology. 1 µM di-8-ANEPPS was loaded for ten minutes in HBSS at room temperature and atmospheric $CO_2$. Coverslips were washed twice in HBSS and transferred to fresh HBSS for imaging. No surfactants were used in the loading (e.g. Pluronic F-127) because their presence worsened cell robustness for whole-cell patch-clamp electrophysiology. All recordings were made with HBSS as an extracellular solution; no cells were kept for more than 30 min after dye loading due to the increasing presence of internalized dye.

Epifluorescence imaging was performed with an inverted Observer.Z1 (Carl Zeiss Microscopy) controlled with µManager 1.4 (Open Imaging) (*Edelstein et al., 2014*). Images were acquired with an Orca Flash 4 Digital CMOS camera (Hamamatsu Corporation; San Jose, CA). Excitation light was provided with a Spectra X light engine (Lumencor, Inc.; Beaverton, OR). Excitation wavelengths were selected with built-in filters in the Spectra X (440/20 bandpass filter for blue and 550/15 bandpass filter for green). Blue-excited images were obtained with an excitation power of 71 mW/$mm^2$ and an exposure time of 50 ms. Green-excited images were obtained with an excitation power of 136 mW/$mm^2$ and an exposure time of 500 ms. Emission light was collected with a 40x magnification oil immersion objective lens using Immersol 518F immersion oil (Zeiss). Fluorescence emission was selected with a 562 nm long pass dichroic mirror and further filtered by a 593/40 bandpass filter (Semrock). Excitation and emission wavelengths were selected to match previous work with this probe as closely as possible (*Zhang et al., 1998*) (current excitation [blue]: 440 ± 10 nm; reported excitation [blue]: 440 ± 15 nm; current excitation [green]: 550 ± 7.5 nm; reported excitation [green]: 530 ± 15 nm; current dichroic: 562 nm long-pass; reported dichroic: 565 nm; current emission: 593 ± 20 nm; reported emission: 570 nm long pass).

## Di-8-ANEPPS data analysis

Single color (e.g. blue excited or green excited) fluorescence images were background subtracted at each pixel before ratios were calculated. The background value was determined from a region of interest near the center of the image that contained no cells and minimal fluorescent debris. Excitation ratios ('R', blue signal divided by green signal, B/G) were then calculated pixelwise from the background subtracted fluorescence images. Pixels with less than 100 arbitrary units of signal in either the blue or the green channel were excluded from analysis and are depicted in black. Regions of interest (ROIs) were manually selected in FIJI to include only area corresponding to the cell membrane. The ratio was averaged across all pixels in a given ROI (similar to the treatment for VF-FLIM, as described in *Figure 1—figure supplement 1*). The ratio values per value of $V_{mem}$ (set by whole cell patch clamp electrophysiology) in *Figure 1—figure supplement 5E,F* are the average of these cell-averaged ratios obtained in 6 or 7 sequential images acquired while the $V_{mem}$ was held at the indicated value.

Where normalized R values are discussed, these values were calculated by dividing the ratio at a given potential (averaged for an ROI as discussed above) by the ratio at 0 mV, as reported previously (*Zhang et al., 1998*). This normalization procedure requires electrode-based calibration for every

individual recording and cannot be stably extended to all cells from a particular cell line. Therefore, it is not analogous to VF-FLIM and is not the point of comparison for voltage resolution.

## Statistical analysis

Mean ± standard error of the mean (SEM) of data is reported throughout the text. Hypothesis testing was performed as indicated with either analysis of variance (ANOVA) followed by appropriate post hoc tests or two-sided, unpaired, unequal variances t-tests. Statistical tests were performed in Python 2 or 3 with the SciPy, pandas and Pingouin (*Vallat, 2018*) packages. Unless otherwise noted, all data shown reflect at least three biological replicates (independent cultures measured on different days). Each of these biological replicates contained between 1 and 5 technical replicates (different samples of cells that were measured on the same day and had been prepared from the same cell stock). For tandem electrophysiology-FLIM measurements, each $\tau_{fl}$-$V_{mem}$ calibration includes at least three biological replicates to capture the variability expected during applications of VF-FLIM. No power analyses were performed before data were collected. Sample sizes throughout the text refer to the total number of cells or cell groups of a given type analyzed across all biological and technical replicates. Cell group identification is discussed in Methods. For experiments where resting membrane potential or resting membrane potential changes are compared to a baseline (*Figures 3–5* and supplements), both control measurements and their physiologically or pharmacologically altered counterparts were recorded on each experimental day. Masking was not used during data collection or analysis.

# Acknowledgements

We thank Holly Aaron and Vadim Degtyar for expert technical assistance and training in the use of FLIM, Prof. John Kuriyan and Dr. Sean Peterson for helpful discussions, and members of the Miller lab for providing VF dyes. FLIM experiments were performed at the CRL Molecular Imaging Center, supported by NSF DBI-0116016. Cell lines were from the UCB Cell Culture Facility which is supported by The University of California Berkeley. FCK-QuasAr2-Citrine was a gift from Adam Cohen (Addgene plasmid # 59172). JLD was supported by an NSF Graduate Research Fellowship. EWM acknowledges support from the Sloan Foundation (FG-2016–6359), March of Dimes (5-FY16-65), and the NIH (R35GM119855).

# Additional information

## Competing interests

Evan W Miller: is listed as an inventor on a patent describing voltage-sensitive fluorophores. This patent (US20170315059) is owned by the Regents of the University of California. The other authors declare that no competing interests exist.

## Funding

| Funder | Grant reference number | Author |
| --- | --- | --- |
| National Science Foundation | GRFP | Julia R Lazzari-Dean |
| National Institutes of Health | R35GM119855 | Evan W Miller |
| Alfred P. Sloan Foundation | FG-2016-6359 | Evan W Miller |
| March of Dimes Foundation | 5-FY-16-65 | Evan W Miller |

The funders had no role in study design, data collection and interpretation, or the decision to submit the work for publication.

## Author contributions

Julia R Lazzari-Dean, Software, Formal analysis, Validation, Investigation, Visualization, Methodology, Writing—original draft, Writing—review and editing; Anneliese MM Gest, Formal analysis, Validation, Investigation, Methodology, Writing—review and editing; Evan W Miller,

Conceptualization, Formal analysis, Supervision, Validation, Methodology, Writing—original draft, Writing—review and editing

## Author ORCIDs
Julia R Lazzari-Dean (iD) https://orcid.org/0000-0003-2971-5379
Evan W Miller (iD) https://orcid.org/0000-0002-6556-7679

## Decision letter and Author response
Decision letter https://doi.org/10.7554/eLife.44522.039
Author response https://doi.org/10.7554/eLife.44522.040

## Additional files

### Supplementary files
• Source code 1. Global analysis of CAESR fluorescence lifetimes.
DOI: https://doi.org/10.7554/eLife.44522.034

• Source code 2. Automated thresholding and cell group identification.
DOI: https://doi.org/10.7554/eLife.44522.035

• Transparent reporting form DOI: https://doi.org/10.7554/eLife.44522.036

### Data availability
All data presented in the manuscript is available in the supporting/supplementary information.

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
