## [Decision Letter]

Thank you for submitting your article "Optical determination of absolute membrane potential" for consideration by *eLife*. Your article has been reviewed by four peer reviewers and the evaluation has been overseen by a Reviewing Editor and Richard Aldrich as the Senior Editor. The following individuals involved in review of your submission have agreed to reveal their identity: Bill Ross (Reviewer #1); Leslie M Loew (Reviewer #2); Bradley Baker (Reviewer #3).

We concluded that it was a potentially interesting and useful method and recommend "revise". We ask that you respond to all of the individual reviewers suggestions and criticisms as well as the items below.

We ask that you carry out additional experiments to compare your method with the method published previously by the Leslie Loew laboratory. In one cell type, determine whether your dye and FLIM give a better estimate of membrane potential than ratio imaging and wide-field imaging with the Les Loew dye.

Lastly, we ask that you be more circumspect in advertising the method. The title, "Optical determination of absolute membrane potential", and wording in the text imply that the method will have millivolt accuracy for every cell and every cell type. The results in the paper do not support such a claim. In addition, the limitations of the method should be more explicitly addressed in the Discussion section. Will the method break down for long recordings because the dye gets internalized? There is a claim that the FLIM measurement is sensitive only to membrane potential, but if you would like to state this you would have to test other variables such as temperature, membrane lipid composition, ion concentration, age of cultured cells etc.

*Reviewer #1:*

This paper describes measurements of membrane potential using FLIM of a class of fluorescent molecules previously shown to respond well to voltage via a PeT mechanism. FLIM of these molecules is much more sensitive to membrane potential changes and more accurately calibrated than similar measurements on GEVIs. The authors show how they calibrate these signals, how they can be used to assay the resting membrane potentials of large numbers of cells, and demonstrate one particular application of this technique to analyze the effect of EGF stimulation of human carcinoma cells. They suggest that this technique could have wide application in situations where it would be helpful to assay slow changes of membrane potential in many cells at the same time.

The measurements appear to be carefully done. I have only a few questions.

1) The paper suggests that all the measurements are made from the surface membrane of the cell, but they do not demonstrate this point. When they calibrate the changes in single cells using voltage clamp, they certainly only record the surface membrane signals. This is partly why the signals are so linear with little difference from cell to cell. But when they look at the resting potential, they cannot be sure there is no signal from internal compartments. They say that, "the vast majority of the fluorescence signal is voltage-sensitive and at the membrane." The confocal images support this claim. But there are no numbers, and confocal images will exaggerate the contribution of surface fluorescence. Since mitochondria and other internal compartments have membranes with different potentials, their contributions must be shown to be small.

2) They claim that the variation from cell to cell is about 20 mV. This appears to be an RMSD evaluation. Figure 1I seems to show that the variation from cell to cell is about 40 mV. These two numbers may be consistent, but in many cases the 40 mV range may the important one to consider. Physiological variations in membrane potential are usually much less than that amount.

*Reviewer #2:*

Overall, this is a careful and thorough study and could have practical applications for screening membrane potential in cell-based assays.

The novelty of this work is diminished, however, since Adam Cohen's lab has already published 2 papers (Hou, Venkatachalam and Cohen, 2014; Brinks, Klein and Cohen, 2015) showing that absolute membrane potential could be measured via time domain recordings. Some comparisons (subsection “VoltageFluor Fluorescence Lifetime Varies Linearly with Membrane Potential”) are made to CAESR from Brinks et al. (2015), which used 2-photon excited fluorescence lifetime measurements to determine absolute membrane potential with lower accuracy than in the current study.

Hou, Venkatachalam and Cohen (2014) reports on 1-photon time-domain measurements of Arch(D95H) and is not fully considered here. There, they report very little cell to cell variation and a sensitivity of a factor of 2 per 100 mV with accuracy of ~10 mV. These measurements are not of fluorescence lifetimes, but rather of voltage-dependent fast photochemical kinetics. Still, this Cohen paper does also show how time domain measurements can allow determination of absolute membrane potential with good accuracy and little cell to cell variability. So together these 2 older papers diminish the novelty of this report.

In subsection “Optical Determination of Resting Membrane Potential Distributions” the authors report ranges of V_mem_ at rest and in the presence of high K^+^. They refer to the calculations using the HGK equation to justify these ranges. However, in the Materials and Methods section, the authors acknowledge that they don't know the appropriate parameters needed by the HGK equation for these cells. Instead, they calculate the HGK equations with many combinations of parameters and claim that the resultant calculated range of values spans the measured range of values. But these ranges are so broad that any cell line would probably fit and really don't prove anything about the validity or accuracy of these resting potential estimates. We are also left with the open question of whether these variations are really due to differences in resting transmembrane potential or some other factor that could alter the lifetime, such as the membrane dipole potential. My lab showed many years ago that dual wavelength ratio imaging of electrochromic VSDs could be used to measure resting potential in single cells (Zhang et al., 1998). But there were some cell to cell differences and even differences in ratio within a single cell that could be attributed to membrane dipole potentials (see: Bedlack et al., 1994; Gross, Bedlack Jr and Loew, 1994). Dipole potentials arise from the particular lipid composition of the membrane and therefore can vary from cell line to cell line or along the surface of a differentiated neuron. I don't see any reason that the PeT mechanism used by VF2.1.Cl wouldn't also be sensitive to the electric field produced by the dipole potential. This could all be checked by doing current clamp measurements of resting potential in the same cell that you measure the fluorescence lifetime.

Exposing the cells to high K^+^ is likely to cause irreversible damage and any assumptions about specific levels of depolarization would be suspect. This might compromise the interpretation of several experiments.

*Reviewer #3:*

I have reviewed the manuscript, “Optical determination of absolute membrane potential”, where the authors employ fluorescent lifetime measurements (FLIM) to quantify membrane potential using a voltage sensing dye. This report is a major improvement over a previous attempt to use FLIM with a genetically encoded voltage indicator. While the information gained by optically measuring membrane potential in response to treatment with EGF shows the power of this technique, I have several concerns that should be addressed before I can completely support publication.

The last paragraph of the Introduction states that this approach can be used in a range of biological contexts. The most glaring concern is what is not in this report. There are no recordings of membrane potential from excitable cells. Why? I see from Figure 1—figure supplement 1 the temporal time scale is in seconds. Is this the problem with recording from neurons? If so, please state this in the main text. If not, please state why.

The last paragraph of the Introduction states a 20-fold improvement in accuracy over previous optical methods yet there is no direct comparison in the manuscript. Please move the CAESR data from the supplementary material (Figure 1—figure supplement 5) into Figure 1. Subsection “VoltageFluor Fluorescence Lifetime Varies Linearly with Membrane Potential” also compares VF2.1.Cl to the genetically encoded voltage indicator giving another reason to include the CAESR data into Figure 1.

In Figure 1G there is a significant range of lifetime measurements in a single cell for the +40 mV membrane potential but it is uniform at +80 mV. Why is that? Is this a common occurrence? I noticed the same thing in the CAESR paper which I contributed to the probe not really working. Perhaps this is a function of lifetime imaging? Or the binning protocol? I think it would also be helpful to have Scheme 2 be Figure 1 to show how the measurement is made and change current Figure 1 to Figure 2.

In subsection “VoltageFluor Fluorescence Lifetime Varies Linearly with Membrane Potential” the authors state that the fractional change in τ(22.4 +/- 0.4%) is in good agreement with the ∆F/F value of 27%. Am I to infer from this that the ∆F/F value is due primarily to a change in lifetime fluorescence? If so, why not use ∆F/F to quantitate membrane potential?

In subsection “VoltageFluor Fluorescence Lifetime Varies Linearly with Membrane Potential” the resolution of membrane potential for a cell is estimated at 4 mV for intra-cellular measurements and 20 mV between different cells. Figure 1H and I show that the slope is more consistent than the absolute value of Lifetime fluorescence. However, this claim is important and should be demonstrated experimentally. Please add a supplementary figure showing 4 mV steps effect on lifetime measurements.

Subsection “VoltageFluor Fluorescence Lifetime Varies Linearly with Membrane Potential” states a resolution of 390 mV. That number does not make sense. Is it supposed to be 39 mV?

Subsection “Evaluation of VF-FLIM across Cell Lines and Culture Conditions” states that despite the variances shown in Figure 2A, all cell lines' membrane potential can be resolved at or under 5 mV. Please show this for CHO cells since it has the most varied slope and MCF-7 cells which showed the highest variance for 0 mV measurement.

*Reviewer #4:*

I felt that the paper promises more than was delivered. The paper claims to have developed "a new method for optically quantifying absolute membrane potential in living cells.....with single cell resolution".

Figure 2I shows that the fluorescence is not a measure of the absolute voltage but that each cell has a different FLIM vs. voltage curve. Thus, calibration with an electrode is needed. Furthermore, the cell to cell differences are not the same from one cell type to another; MCF-7 cells displayed greater variability than other cell lines tested (Figure 2B). Thus, calibration for a new cell type will need to include measuring the FLIM vs. voltage response from many cells.

Many images show blobs (Figure 1E and F, Figure 2C, Figure 3, Figure 4, Figure 5B). Sometimes the blobs seem voltage dependent, sometimes not. I would presume that the blobs are the result of non-specific staining. This subject was not discussed. Were the presented images selected for relatively good membrane staining?

In Figure 4A and Figure 5B different parts of the cell membrane appear to have different voltage responses and these response differences do not seem to be stochastic. This result does not fit with expected membrane voltage uniformity for small cells.

Drawbacks of the method are not discussed. The time resolution seems relatively slow. Will the method will be applicable to preparations with substantial light scattering? How will it work in three dimensional preparations? The differences from cell type to cell type will require calibration for each cell type and perhaps for each developmental age of each cell type.

[Editors' note: further revisions were requested prior to acceptance, as described below.]

Thank you for resubmitting your work entitled "Optical estimation of absolute membrane potential using fluorescence lifetime imaging" for further consideration at *eLife*. Your revised article has been favorably evaluated by Richard Aldrich (Senior Editor), and Lawrence Cohen (Reviewing Editor), and three other reviewers. The following reviewer has agreed to share their identity: Leslie M Loew (Reviewer #2).

The manuscript has been improved but there are some remaining issues that need to be addressed before acceptance, as outlined below:

*Reviewer #1:*

I am satisfied with the revision.

*Reviewer #2:*

The authors have significantly revised and thereby strengthened the paper. In particular, they did scale back on some of the over-enthusiastic claims for their method. They also did a good job comparing their method to previous lifetime-based and dual wavelength ratio-based approaches. The new method with the VF dyes is indeed more sensitive and seems to have less cell to cell variability for estimating absolute V_mem_. I appreciate the clearer explanation of how the GHK equation was used to understand the high K^+^ experiments. I still would have preferred some current clamp electrophysiology experiments to validate the resting potential determinations, but this is not essential. There are still 2 areas related to my own previous work where there is need for some clarification:

1) In comparing the ratiometric electrochromic dyes in the Introduction, they state: "Although they benefit from simpler loading procedures, signals from electrochromic styryl dyes display a strong dependence on local membrane properties other than transmembrane potential, reducing the accuracy of V_mem_ determinations (Gross et al., 1994; Montana et al., 1989; Zhang et al., 1998)". This is correct when examining different cell lines or cells in different states of differentiation, where the lipid composition, for example, can affect the ratio. But saying this in the Introduction appears to suggest that the new method that is about to be described does not have this problem; it likely does and even shows differences within the same cell line. And, actually, examination of Figure 2 in Zhang et al. (1998) shows that the cell to cell variation for the normalized ratiometric approach is remarkably small for the 40 cells examined. Figure 4 in Zhang et al. (1998) shows remarkably little variation in V_rest_ for undifferentiated neuroblastoma cells; the small variation in V_rest_ for differentiated cells, may be attributed to different degrees of differentiation.

2) In discussing the influence of membrane dipole potential, the authors misunderstand some of the studies from my lab, subsection “Resolution of VF-FLIM: Voltage, Space, and Time”:

"Relative to di-8-ANEPPS, where this effect was documented (Gross et al., 1994; Zhang et al., 1998), VF-FLIM displays less cell to cell variability, suggesting reduced dependence on the membrane dipole potential. The reason for this is unclear, as both sensors putatively detect V_mem_ from within the plasma membrane (Loew et al., 1979; Miller et al., 2012)." Our studies deliberately sought to establish the sensitivity of d-8-ANEPPS to dipole potential by systematically measuring ratios with different lipid compositions in lipid vesicles and by adding or depleting cholesterol in cell membranes. As a side benefit, these studies showed that membrane composition had to be considered when using dual wavelength ratio measurements to determine absolute V_mem_. Until the authors do a deliberate investigation of this effect on FLIM of their probes, I don't think they can say that FLIM of the VF dyes is less sensitive to membrane composition.

*Reviewer #3:*

No further major comments.

*Reviewer #4:*

The revision is greatly improved. However, there are several areas where the paper remains overstated.

In the Abstract add the words "in culture" after the words "single cell resolution".

The first paragraph of the Introduction leads the reader to think that the paper is about signals that can be measured rapidly and can be measured in complex tissues even though the reported measurements have a time resolution that is ~four orders of magnitude slower than presently available from other methods and the measurements are only from single cells in culture. This needs toning down. The timing issue would be clearer for the reader if the table in subsection "Acquisition Time and Effective Pixel Size in Lifetime Data”" was in the main body of the paper.

A discussion of the difficulties of applying this method to other applications should be added to the Discussion section.

The Discussion section notes that faster apparatus is available. How much faster? One order?

Please discuss the possible explanations for "between regions" in the following sentence in the Discussion section: "Intriguingly, there are differences in lifetime within some cells in VF-FLIM images, both at the pixel to pixel level and between regions of the cell membrane."

---

## [Author Response]

We ask that you carry out additional experiments to compare your method with the method published previously by the Leslie Loew laboratory. In one cell type, determine whether your dye and FLIM give a better estimate of membrane potential than ratio imaging and wide-field imaging with the Les Loew dye.

First and foremost, we would like to apologize for omitting any mention of the ratiometric voltage sensors developed by Leslie Loew in our original submission. We thank the reviewers for pointing out this oversight and giving us an opportunity to address this.

We performed wide-field epifluorescence imaging of di-8-ANEPPS excitation ratios as requested (Figure 1—figure supplement 5) and compared the voltage resolution with VF-FLIM. We find that VF-FLIM outperforms di-8-ANEPPS in voltage resolution by 8-fold. With di-8-ANEPPS in HEK293T, we record a membrane potential resolution of 150 mV (“inter cell,” see Materials and Methods section); with VF-FLIM this resolution is 19 mV. We have incorporated description of these data in the subsection “VoltageFluor Fluorescence Lifetime Varies Linearly with Membrane Potential”, as well as in subsection “Resolution of VF-FLIM: Voltage, Space, and Time”.

Imaging and data processing for di-8-ANEPPS are described in the subsection “Di-8-ANEPPS Ratio-based Imaging”; we matched experimental conditions described by Zhang et al. (1998) as closely as possible with our microscope.

Notably, di-8-ANEPPS has 3-fold better V_mem_ resolution than the GEVI CAESR, our previous point of comparison. We appreciate this insightful experiment suggestion from the reviewers, as inclusion of the di-8-ANEPPS data has allowed us to more comprehensively compare VF-FLIM with the previous state-of-the-art in optical V_mem_ determinations.

Lastly, we ask that you be more circumspect in advertising the method. The title, "Optical determination of absolute membrane potential", and wording in the text imply that the method will have millivolt accuracy for every cell and every cell type. The results in the paper do not support such a claim.

We appreciate this feedback about representation of the VF-FLIM method, as well as the opportunity to offer clarification. We were surprised to hear that the reviewers got the impression of 1 mV accuracy in all cell types from the text. This was not our intent.

In our original submission, we report the accuracy as ~20 mV for absolute V_mem_ (“inter-cell” comparisons) and ~5 mV for absolute V_mem_ changes (“intra-cell” comparison). In the original submission, the values for the 5 tested cell lines can be found in subsection “VoltageFluor Fluorescence Lifetime Varies Linearly with Membrane Potential” and in Figure 2—source data 1. These data are still present in our revised manuscript. In our initial submission, we also show all of our single-cell electrophysiology results for all cell types/lines (Figure 1F-J, Figure 2, and Figure 2—figure supplement 1), which gives the reader a transparent view of the lifetime data.

We believe that some of the misunderstanding may arise from the term “absolute V_mem_.” Absolute V_mem_, in our usage, does not imply any particular level of voltage resolution. The term absolute V_mem_ has been used previously in the literature to describe optical V_mem_ recordings that can be interpreted as V_mem_ rather than arbitrary fluorescence units (Hou et al., 2014 and Brinks et al., 2015). We have clarified the meaning of this term in the Introduction and added additional mention of the voltage resolution.

Regarding the advertising of the method, we are excited about VF-FLIM and the advantages that it brings to the field. We had no intention of misleading readers. We have therefore carefully reviewed the entire manuscript in detail and identified areas with ambiguous, or potentially confusing, text about the performance of VF-FLIM. We have made the following updates in this revised version, which we believe provides a transparent and well-rounded description of VF-FLIM.

1) We restructured and expanded the early paragraphs of subsection “Resolution of VF-FLIM: Voltage, Space, and Time” to address the different dimensions of VF-FLIM’s resolution (voltage, space, and time).

2) We elaborated upon factors other than V_mem_ that can affect lifetime (Introduction, subsection “Resolution of VF-FLIM: Voltage, Space, and Time”). We also added a reference to an excellent review that discusses this topic in depth (Berezin and Achilefu, 2010).

3) We recognize that the second-timescale acquisitions of VF-FLIM make certain applications challenging (especially those in excitable cells), so we have added this information to the Introduction in anticipation of questions from readers.

4) We have edited the language describing the scope of biological applications tested throughout the text, as all of the data presented here are in cultured cell lines. We have changed the less specific terms “cell type” and “biological context” to “cell line” when we refer to the experiments we performed.

5) We have added specific commentary in the Discussion about the requirement for initial calibration (subsection “Resolution of VF-FLIM: Voltage, Space, and Time”). We believe the importance of calibration is clear from the abundance of voltage clamp data throughout the manuscript, but we want to be sure that readers are bearing this in mind when they consider other possible applications of VF-FLIM.

6) We added a new Scheme 2 to the Materials and methods section, which graphically describes the voltage resolution (reported as root mean square deviation, RMSD) calculations and supplements existing mathematical description in the text. In addition, we now state that the reported V_mem_ resolutions are RMSD-based when we mention them in the main text.

7) We agree that the title would benefit from additional specificity, so we propose “Optical estimation of absolute membrane potential using fluorescence lifetime imaging”.

In addition, the limitations of the method should be more explicitly addressed in the Discussion section. Will the method break down for long recordings because the dye gets internalized? There is a claim that the FLIM measurement is sensitive only to membrane potential, but if you would like to state this you would have to test other variables such as temperature, membrane lipid composition, ion concentration, age of cultured cells etc.

We appreciate the invitation to discuss the nuances of FLIM further, and we have amended the Discussion section to reflect the reviewers’ points. We are not able to find a location in our original submission where we state that FLIM is only dependent on V_mem_, and we certainly did not mean to imply it. Our best guess is that the confusion arose in the Introduction, where we discuss how FLIM avoids drawbacks of intensity imaging. We have modified this text to make it clear that FLIM is not a panacea.

In our original submission, we included detailed analysis of the dependence of VoltageFluor lifetime on certain factors other than V_mem_: (a) dye concentration (subsection “VoltageFluor Fluorescence Lifetime Varies Linearly with Membrane Potential”, subsection “Evaluation of VF-FLIM across Cell Lines and Culture Conditions”, Figure 1—figure supplement 2 and Figure 2—figure supplement 4); (b) cell groups vs. individual cells, (c) and in culture conditions involving removal of serum (subsection “Evaluation of VF-FLIM across Cell Lines and Culture Conditions”,, Figure 2—figure supplement 3). We did not see substantial changes in VF-FLIM under these different growth conditions.

This information is key to readers who wish to implement VF-FLIM in their own laboratories.

Nevertheless, we do agree with the reviewers that the previous discussion did not provide sufficient commentary on potential pitfalls of the approach. We have included more discussion of this (Introduction, subsection “Resolution of VF-FLIM: Voltage, Space, and Time”). We also included additional specificity about FLIM’s time resolution and discussed the variability in lifetime between pixels in images, which is most likely noise from the biexponential fitting process (subsection “Resolution of VF-FLIM: Voltage, Space, and Time”).

We would like to specifically address some of the potential confounding factors mentioned by the reviewers above. We did not test effects of temperature and ion concentration because these variables are tightly constrained by the biological system of interest. The cellular toxicity from dramatic shifts in temperature or ionic strength would likely supersede effects on the lifetime-V_mem_ calibration and make electrophysiology challenging. However, we now mention these potential confounding factors in subsection “Resolution of VF-FLIM: Voltage, Space, and Time”. Regarding age and condition of cultured cells, we used cells ranging from freshly thawed to passage 25 (as documented in subsection “Cell culture”) and did not see any obvious trends with passage number.

The effects of membrane composition on voltage sensors are highly nuanced and interesting. We had previously omitted this from the Discussion because we did not want to speculate excessively as to which factors limit the resolution in VF-FLIM recordings. Membrane composition is inherently complex and difficult to assess comprehensively. While membrane compositional differences between cells perhaps contributes to the noise in VF-FLIM, we have not conclusively shown that majority of the noise in our V_mem_ measurements is the result of membrane composition. We agree, though, that knowledge of this possible confounding factor is important for readers and potential future users of the method, so we now mention this in subsection “Resolution of VF-FLIM: Voltage, Space, and Time”.

Regarding stability of VF2.1.Cl stain on the plasma membrane, we retain good membrane staining and see relatively little internalized dye, even following two hours of incubation of stained cells at 37°C. We believe that the high percentage of active, correctly localized VoltageFluor is one source of the V_mem_ resolution improvements over other optical strategies. Some punctate fluorescence spots do accumulate, but they are generally separable from the membrane fluorescence and do not interfere with recordings. These puncta are presumably attributable to internalized dye. We show that these puncta have little effect in Author response image 1, where we analyze VF-FLIM data using pixels from the cell’s interior in addition to pixels from the plasma membrane. Furthermore, in previous work, we have shown that the vast majority of the VF signal is located at the membrane (through quenching experiments with Trypan Blue in Grenier et al., 2019).

Reviewer #1:[…]1) The paper suggests that all the measurements are made from the surface membrane of the cell, but they do not demonstrate this point. When they calibrate the changes in single cells using voltage clamp, they certainly only record the surface membrane signals. This is partly why the signals are so linear with little difference from cell to cell. But when they look at the resting potential, they cannot be sure there is no signal from internal compartments. They say that, "the vast majority of the fluorescence signal is voltage-sensitive and at the membrane." The confocal images support this claim. But there are no numbers, and confocal images will exaggerate the contribution of surface fluorescence. Since mitochondria and other internal compartments have membranes with different potentials, their contributions must be shown to be small.

The specificity of the stain is an important point; we have looked at this in a few different ways. First, although the VF-FLIM data shown here were taken on a laser scanning confocal, we use a relatively large pinhole (2.5-3.5 Airy units, which corresponds to a ~2.5 µm optical section). We choose to sacrifice optical sectioning for the sake of photon count, as exponential fitting of lifetime data requires large numbers of photons. We have added this information to the subsection “Acquisition Time and Effective Pixel Size in Lifetime Data“. Second, in previously published work (Grenier et al., 2019), our lab determined that >80% of the signal from VF2.1.Cl at the cell membrane can be quenched by the addition of extracellular trypan blue. This experiment was performed under epifluorescence conditions, and it is a conservative estimate of the percentage of plasma membrane signal, as it is unlikely that trypan blue would completely quench all plasma membrane associated dye.

To answer your question for VF-FLIM specifically, we quantified the contribution of internal signal in one of our lifetime datasets (see Author response image 1). We processed the HEK293T lifetime-V_mem_ electrophysiology calibration data with different regions of interest (ROIs, Author response image 1) to explore the effects of internal signal. In this analysis, we documented the effect of including all interior pixels (“whole cell”) ROIs instead of just selecting membrane pixels (Author response image 1). Inclusion of this interior signal produces a small decrease in both the average slope (3.2 ± 0.1 ps/mV including the interior versus 3.50 ± 0.08 ps/mV with plasma membrane only, p = 0.033; data are mean ± SEM and p values are from a paired Student’s t-test) and average 0 mV lifetime (1.74 ± 0.02 ns including the interior versus 1.77 ± 0.02 ps/mV with plasma membrane only, p = 0.26). Using one slope and intercept estimate as opposed to the other results in V_mem_ values that differ by less than 10 mV in the range -100 to 0 mV.

Because we have reasonable spatial resolution in our images, it is feasible to select an ROI that largely encompasses the plasma membrane. Selection of different, smaller ROIs does not introduce systematic error into the measured lifetimes, which indicates that the vast majority of signal in all areas of the cell results from plasma membrane-localized VF2.1.Cl.

Taken together, these analyses reveal that the internal signal from VF2.1.Cl stain is minimal and does not interfere with VF-FLIM measurements.

**Author response image 1. respfig1:** Effect of internal signal on HEK293T VF-FLIM calibrations. (**A**) Illustration of different regions of interest (ROIs) used for processing the data. All VF-FLIM analysis in other parts of the manuscript was performed with “membrane” ROIs. Whole cell ROIs include all interior pixels that are above the threshold for lifetime fitting (300 peak photons, see Materials and Methods section). (**B**) Effect of internal signal on the sensitivity (slope) of the HEK293T τ_fl_-V_mem_ data shown in Figure 1 of the main text. Gray points are results from individual cells; aggregated data are shown as mean ± SEM of n=17 cells. (**C**) Effect of internal signal on the 0 mV lifetime (y-intercept) of the lifetime-V_mem_ calibration. P values for (**B**) and (**C**) were determined using paired Student’s t tests.

2) They claim that the variation from cell to cell is about 20 mV. This appears to be an RMSD evaluation. Figure 1I seems to show that the variation from cell to cell is about 40 mV. These two numbers may be consistent, but in many cases the 40 mV range may the important one to consider. Physiological variations in membrane potential are usually much less than that amount.

Yes, the voltage resolution calculations are based on an RMSD. We now clearly state this in the main text (Introduction, subsection “VoltageFluor Fluorescence Lifetime Varies Linearly with Membrane Potential”), in addition to our description in the subsection “Resolution of VF-FLIM Voltage Determination”. A similar calculation was also performed in previous work with GEVIs (Hou et al., 2014) to determine voltage accuracy of their approach. We agree that the spread, not the RMSD, estimates the maximal potential error in an individual measurement. Given the throughput of VF-FLIM, we envision that the vast majority of users will be repeating their measurements multiple times, in which case measures of standard deviation are more informative. To illustrate the resolution calculations more clearly, we have expanded upon our description of this (subsection “Resolution of VF-FLIM Voltage Determination”) and added a new Scheme 2 to the Materials and Methods section, which graphically depicts the RMSD calculation as well as the spread of the HEK293T data. We also report all of the single cell lifetime-membrane potential standard curves, which show the full raw datasets and thereby illustrate spread (Figure 1, Figure 2, Figure 2—figure supplement 1).

Reviewer #2:[…]The novelty of this work is diminished, however, since Adam Cohen's lab has already published 2 papers (Hou, Venkatachalam and Cohen, 2014; Brinks, Klein and Cohen, 2015) showing that absolute membrane potential could be measured via time domain recordings. Some comparisons (subsection “VoltageFluor Fluorescence Lifetime Varies Linearly with Membrane Potential”) are made to CAESR from Brinks et al. (2015), which used 2-photon excited fluorescence lifetime measurements to determine absolute membrane potential with lower accuracy than in the current study.

We recognize and appreciate the contribution of Adam Cohen’s lab in this area. We demonstrate that the use of VoltageFluors rather than CAESR with FLIM greatly improves the voltage resolution of absolute membrane potential recordings (Figure 1—figure supplement 4); this voltage resolution improvement makes it possible to map biological V_mem_ signals optically.

Hou, Venkatachalam and Cohen (2014) reports on 1-photon time-domain measurements of Arch(D95H) and is not fully considered here. There, they report very little cell to cell variation and a sensitivity of a factor of 2 per 100 mV with accuracy of ~10 mV. These measurements are not of fluorescence lifetimes, but rather of voltage-dependent fast photochemical kinetics. Still, this Cohen paper does also show how time domain measurements can allow determination of absolute membrane potential with good accuracy and little cell to cell variability. So together these 2 older papers diminish the novelty of this report.

The use of fast photochemical kinetics with Arch(D95H) to record membrane potential is indeed an interesting strategy. We were unable to compare VF-FLIM directly to this approach because we lack instrumentation suitable for fast photochemical kinetics. We have amended the Introduction to state this more explicitly. Unlike the instruments for fast photochemical kinetics used in Hou, et al. (2014), lifetime instruments are commercially available from a number of different suppliers. Therefore, we believe that FLIM may be a more accessible strategy for general use in absolute membrane potential recording. Additionally, the presence of non-specific intracellular fluorescence is a major concern with GEVI-based absolute V_mem_ strategies. To remedy this, GEVI-based images (including those in Hou et al., 2014) are often masked or processed to include only pixels responding to an external stimulus in the analysis. Such image processing becomes challenging when there are no fast dynamics in V_mem_, as is the case with resting membrane potential or slow changes. Thus, we believe that VF-FLIM represents an important step forward in feasibility, resolution, and ease of interpretation of absolute V_mem_ data.

In subsection “Optical Determination of Resting Membrane Potential Distributions” the authors report ranges of V_mem_ at rest and in the presence of high K^+^. They refer to the calculations using the HGK equation to justify these ranges. However, in the Materials and Methods section, the authors acknowledge that they don't know the appropriate parameters needed by the HGK equation for these cells. Instead, they calculate the HGK equations with many combinations of parameters and claim that the resultant calculated range of values spans the measured range of values. But these ranges are so broad that any cell line would probably fit and really don't prove anything about the validity or accuracy of these resting potential estimates.

We do not mean to imply that the Goldman equation with estimated permeabilities and ion concentration is in any way an accurate prediction of membrane potential. To provide direct support for our optically recorded resting membrane potentials, we cite literature examples of whole cell patch clamp recordings of resting membrane potential for each cell line in Figure 3—source data 1. Because high K^+^ electrophysiology recordings are uncommon in the literature, our intention with the Goldman equation calculations was to provide some context for the high K^+^ VF-FLIM measurements (Figure 3, Figure 3—figure supplement 1 and Figure 3—figure supplement 2). It was initially counter-intuitive to us that the 120 mM K^+^ treatment does not depolarize all cells to 0 mV, so we wanted to provide a basis for the ranges we observe. Indeed, with the Goldman equation, we predict that variations in the intracellular ion concentrations and permeabilities can produce a range of V_mem_ values at 120 mM external K^+^. We have modified the text in the Results and Materials and Methods sections to make this clear (subsection “Optical Determination of Resting Membrane Potential Distributions”, Materials and methods section).

We are also left with the open question of whether these variations are really due to differences in resting transmembrane potential or some other factor that could alter the lifetime, such as the membrane dipole potential. My lab showed many years ago that dual wavelength ratio imaging of electrochromic VSDs could be used to measure resting potential in single cells (Zhang et al., 1998). But there were some cell to cell differences and even differences in ratio within a single cell that could be attributed to membrane dipole potentials (see: Bedlack et al., 1994; Gross, Bedlack Jr and Loew, 1994.). Dipole potentials arise from the particular lipid composition of the membrane and therefore can vary from cell line to cell line or along the surface of a differentiated neuron. I don't see any reason that the PeT mechanism used by VF2.1.Cl wouldn't also be sensitive to the electric field produced by the dipole potential. This could all be checked by doing current clamp measurements of resting potential in the same cell that you measure the fluorescence lifetime.

We apologize for not initially referencing work using di-8-ANEPPS to record absolute V_mem_ optically; this was an oversight on our part. We address first our quantification of non-V_mem_ factors, followed by discussion of the effects of membrane dipole potential.

The RMSD-based resolution calculations we performed quantify the effects of variations in any non-V_mem_ factors on the lifetime measurement. Anything that is not V_mem_ but changes lifetime (including membrane composition) would be seen as a higher RMSD between the theoretical V_mem_ from electrophysiology and the value of our lifetime measurement. We have modified the text in the subsection “Resolution of VF-FLIM Voltage Determination” and added a Scheme 2 to clarify our resolution calculations.

The RMSD-based resolution that we calculate gets at the same idea as the simultaneous current clamp and fluorescence ratio experiments described with di-8-ANEPPS ratios (Zhang et al., 1998). In both cases, a V_mem_ recorded with electrophysiology is compared to a simultaneous optical estimate of V_mem_ on that cell, and any difference between the two is presumed to arise from artifacts in the lifetime measurement. The difference between our approach here and the published one is the use of voltage clamp versus current clamp mode in the electrophysiology. This change shouldn’t make a difference in the case of HEK293T, which can be voltage clamped accurately over a relatively large range of V_mem_.

Regarding the effects of the membrane dipole potential absolute V_mem_ recordings, it is intriguing to compare our results with di-8-ANEPPS to those with VF2.1.Cl. We have added this comparison to the manuscript, both as a figure supplement (Figure 1—figure supplement 5) and as additional text (subsection “VoltageFluor Fluorescence Lifetime Varies Linearly with Membrane Potential”, subsection “Resolution of VF-FLIM: Voltage, Space, and Time”). In these experiments, we observed that the V_mem_ resolution of VF-FLIM is ~8 fold better than the V_mem_ resolution of di-8-ANEPPS ratios (19 mV inter-cell resolution for VF-FLIM vs. 150 mV for di-8-ANEPPS in HEK293T). Put another way, VF-FLIM is 8-fold less sensitive to factors other than V_mem_ than di-8-ANEPPS based ratio measurements. This result is a bit surprising: both probes sense V_mem_ from within the hydrophobic core of the membrane and therefore could be affected by the membrane dipole potential.

Two possibilities seem likely to us: either (1) VF2.1.Cl is less sensitive to the membrane dipole potential than di-8-ANEPPS or (2) some of the variability in di-8-ANEPPS measurements is from factors other than the membrane dipole potential. Both of these explanations are plausible, and both may play some role. The fluorescein chromophore in VF2.1.Cl is putatively not in the plasma membrane (Kulkarni et al., 2017), which may partially insulate VFs from dipole potential effects. We also did not see differences in the lifetime-V_mem_ calibration in serum-starved versus normally cultured A431 cells (Figure 2—figure supplement 3). Such a growth perturbation is likely to have some effect on membrane composition and dipole potential. Towards point (2), in our hands, the VF2.1.Cl stain appears to be more stably and specifically localized to the plasma membrane than the di-8-ANEPPS stain, so some variability in di-8-ANEPPS signals may be from probe that is adhered to the coverslip or internalized in the immediate vicinity of the membrane (and is therefore not spatially separable, even with a membrane-only ROI).

In sum, the inter-cell V_mem_ resolution values include contributions from all non-V_mem_ factors, including membrane composition effects. We agree that it is important for readers to know about possible effects of membrane composition, so we now address this, as well as other potential confounding factors in FLIM measurement, in the subsection “Resolution of VF-FLIM: Voltage, Space, and Time”.

Exposing the cells to high K^+^ is likely to cause irreversible damage and any assumptions about specific levels of depolarization would be suspect. This might compromise the interpretation of several experiments.

We agree with this assessment of high K^+^ toxicity. We largely use the high K^+^ disruption as an endpoint perturbation to get a sense of the shift in resting membrane potential distributions (Figure 3 and supplements). We also used high K^+^ as one piece of evidence to suggest that the current was K^+^ mediated in the pharmacology studies with EGF treatment of A431 cells (Figure 5A, and Figure 5—figure supplement 1B). In the absence of any other data, we agree this result would not be conclusive. However, we also see loss of the hyperpolarizing response in A431 cells when Ca^2+^ activated K^+^ channels are blocked more specifically with CTX and TRAM-34 (Figure 5A, and Figure 5—figure supplement 1D,E), as well as when intracellular Ca^2+^ is perturbed by addition of the intracellular Ca^2+^ chelator, BAPTA-AM (Figure 5A, and Figure 5—figure supplement 1G). The interpretation that the hyperpolarizing step is mediated by K_Ca_3.1 does not hinge on the high K^+^ perturbation alone.

Reviewer #3:[…]The last paragraph of the Introduction states that this approach can be used in a range of biological contexts. The most glaring concern is what is not in this report. There are no recordings of membrane potential from excitable cells. Why? I see from Figure 1—figure supplement 1 the temporal time scale is in seconds. Is this the problem with recording from neurons? If so, please state this in the main text. If not, please state why.

“Biological contexts” is an ambiguous term. We have edited the Introduction to read “cell lines,” which is the context in which we have demonstrated VF-FLIM’s function in this work.

We focused on non-excitable cells in this work, as our current research interests lie in slower changes in resting membrane potential present in these systems. That said, the time resolution of the FLIM approach does make recording of action potentials from neurons challenging. In our original manuscript, we outlined this in Figure 1—source data 1. We now have incorporated this point more prominently in both the Introduction and subsection “Resolution of VF-FLIM: Voltage, Space, and Time”. While it will probably be always be challenging to accurately record lifetimes at the ~1 kHz frame rate required for neuronal activity imaging, VF-FLIM could likely be used to analyze resting membrane potentials of neurons or other excitable tissue. However, the VoltageFluor used in this text (VF2.1.Cl) labels all plasma membranes, not just those of particular cells of interest. In samples with complex processes and intertwined membranes, attribution of signal to a particular cell is challenging. In the Discussion section, we mention that the use of VF-FLIM in more complex samples and tissues may be most successful in conjunction cell-targeted VoltageFluors, which is ongoing work in our laboratory (e.g. Liu et al., 2017; Grenieret al., 2019).

The last paragraph of the Introduction states a 20-fold improvement in accuracy over previous optical methods yet there is no direct comparison in the manuscript. Please move the CAESR data from the supplementary material (Figure 1—figure supplement 5) into Figure 1. Subsection “VoltageFluor Fluorescence Lifetime Varies Linearly with Membrane Potential” also compares VF2.1.Cl to the genetically encoded voltage indicator giving another reason to include the CAESR data into Figure 1.

We have now added another direct comparison to VF-FLIM, this time an assessment of the ratio-based VSD, di-8-ANEPPS, to Figure 1—figure supplement 4. Together with the comparison to CAESR, which should be readily accessible to readers in *eLife*’s online format, we think that this does a nice job of comparing alternative methods for estimation of absolute V_mem_. We now include additional discussion of these comparisons in the main text (subsection “VoltageFluor Fluorescence Lifetime Varies Linearly with Membrane Potential”, subsection “VoltageFluor Fluorescence Lifetime Varies Linearly with Membrane Potential”).

In Figure 1G there is a significant range of lifetime measurements in a single cell for the +40 mV membrane potential but it is uniform at +80 mV. Why is that? Is this a common occurrence? I noticed the same thing in the CAESR paper which I contributed to the probe not really working. Perhaps this is a function of lifetime imaging? Or the binning protocol? I think it would also be helpful to have Scheme 2 be Figure 1 to show how the measurement is made and change current Figure 1 to Figure 2.

The pixel to pixel differences in lifetime are interesting, and we didn’t discuss this sufficiently in the original manuscript. We have added text to subsection “VoltageFluor Fluorescence Lifetime Varies Linearly with Membrane Potential” to address this for the readers. We also moved the former Scheme 2 about how the FLIM measurements are made to be Figure 1—figure supplement 1 so that it is more readily accessible, especially in the *eLife* online format.

We believe the pixel to pixel variability is primarily noise in the measurement, as the V_mem_ of a small, approximately spherical cell should be uniform. The differences between pixels occur at a constant rate at different potentials, but the rainbow color scale perhaps highlights the differences in some color ranges more than others. The pixel to pixel variability seems to be random noise in the measurement. The most likely source of this noise is the fitting of lifetime data to a biexponential model. Lifetime values at the pixel level are determined from 20 to 100-fold fewer photons than the lifetime value for the region of interest as a whole. This reduction in signal leads to an increase in noise at single pixels versus the region of interest overall. We do not interpret pixel to pixel differences as V_mem_ differences in our studies. In principle, more photons could be acquired per pixel to decrease this variability and enable subcellular interpretation, although it would result in longer acquisition times.

In subsection “VoltageFluor Fluorescence Lifetime Varies Linearly with Membrane Potential” the authors state that the fractional change in τ(22.4 +/- 0.4%) is in good agreement with the ∆F/F value of 27%. Am I to infer from this that the ∆F/F value is due primarily to a change in lifetime fluorescence? If so, why not use ∆F/F to quantitate membrane potential?

ΔF/F is still fundamentally a fluorescence intensity measurement, so it suffers from the fluorescence intensity artifacts that we discuss in the Introduction. ΔF/F therefore cannot be robustly calibrated as absolute V_mem_. A deeper discussion of this topic, as well as how lifetime avoids many of these issues, is available in a review we cite in our Introduction (Yellen and Mongeon, 2015). We also add a citation to an excellent review by Berezin and Achilefu (2010), that provides additional context and clarification for using FLIM (vs. intensity-only measurements) in biological contexts.

We mention the agreement with ΔF/F because it is relevant for our hypothesis that VoltageFluor-type dyes sense voltage via a photoinduced electron transfer (PeT) mechanism. For PeT-based sensors, fluorescence lifetime and fluorescence intensity should be complementary readouts of the same PeT process, although each is useful for different biological applications. The similarity in the ΔF/F and Δτ/τ is consistent with our conclusion that the lifetime recordings represent voltage sensing via a PeT-based mechanism (but, of course, does not entirely rule out other mechanisms). We have modified subsection “VoltageFluor Fluorescence Lifetime Varies Linearly with Membrane Potential” to clarify this point.

In subsection “VoltageFluor Fluorescence Lifetime Varies Linearly with Membrane Potential” the resolution of membrane potential for a cell is estimated at 4 mV for intra-cellular measurements and 20 mV between different cells. Figure 1H and I show that the slope is more consistent than the absolute value of Lifetime fluorescence. However, this claim is important and should be demonstrated experimentally. Please add a supplementary figure showing 4 mV steps effect on lifetime measurements.

The values of our voltage resolution are root mean square deviations (RMSD) measurements, which give a sense of the ‘typical’ amount of error in a measurement (or, put another way, give a sense of our limit of detection). Determination of this RMSD does not require voltage clamping a cell every 4 mV, although the RMSD we determine indicates that statistically significant differences would be seen when comparing a few patches at, say -56 mV, with measurements at, say -60 mV. To clarify where our resolution estimates are coming from, we have expanded/modified our discussion of accuracy and the RMSD calculation (subsection “Resolution of VF-FLIM Voltage Determination”) and added a graphical explanation of this (Scheme 2).

Subsection “VoltageFluor Fluorescence Lifetime Varies Linearly with Membrane Potential” states a resolution of 390 mV. That number does not make sense. Is it supposed to be 39 mV?

No. We obtained the value of 390 mV for inter cellular voltage resolution for CAESR by using the same RMSD calculations that we applied to get the resolution of VF-FLIM. We have now applied the same methods to an analysis of di-8-ANEPPS, which has a resolution of 150 mV. The VF-FLIM approach, with an inter-cellular resolution of 19 mV (RMSD), represents approximately 8- to 19-fold improvement in resolution in this regard, which allows us to use VF-FLIM to explore biologically relevant differences in V_mem_, which are on the order of tens of millivolts, rather than hundreds of millivolts.

We do have a small correction to make: the value of 390 mV should have read 370 mV because of a minor error in the calculation of all RMSDs (for both VF-FLIM and CAESR), causing all values for all approaches to be slightly too high. All numbers are now updated and correct as written (inter-cell resolution in HEK293T of 370 mV for CAESR, 150 mV for di-8-ANEPPS, and 19 mV for VF-FLIM).

Subsection “Evaluation of VF-FLIM across Cell Lines and Culture Conditions” states that despite the variances shown in Figure 2A, all cell lines' membrane potential can be resolved at or under 5 mV. Please show this for CHO cells since it has the most varied slope and MCF-7 cells which showed the highest variance for 0 mV measurement.

To clarify – in our initially submitted manuscript, we do not state that we are able to resolve absolute V_mem_ at or under 5 mV. The resolution for this calculation is 10-23 mV (inter-cell error), depending on the cell line. We are able to resolve absolute membrane potential changes with accuracy at or under 5 mV (intra-cell error). These values are stated in the text (subsection “VoltageFluor Fluorescence Lifetime Varies Linearly with Membrane Potential”, subsection “Evaluation of VF-FLIM across Cell Lines and Culture Conditions”) and are tabulated in Figure 2—source data 1.

Our resolution estimates come from root mean square deviation calculations (RMSD) between the V_mem_ values from electrophysiology and the optically determined V_mem_ values. Resolution is calculated per cell line, so it already takes into account variations in slope and y intercept. We have added a new Scheme 2 to the Materials and methods section that visually illustrates how this calculation is performed as a supplement to the existing mathematical description in the Materials and methods section.

Reviewer #4:I felt that the paper promises more than was delivered. The paper claims to have developed "a new method for optically quantifying absolute membrane potential in living cells.....with single cell resolution".Figure 2I shows that the fluorescence is not a measure of the absolute voltage but that each cell has a different FLIM vs. voltage curve. Thus, calibration with an electrode is needed. Furthermore, the cell to cell differences are not the same from one cell type to another; MCF-7 cells displayed greater variability than other cell lines tested (Figure 2B). Thus, calibration for a new cell type will need to include measuring the FLIM vs. voltage response from many cells.

Fluorescence lifetime of VoltageFluors is a reporter for absolute voltage; it indeed is not voltage. However, the requirement for calibration exists with essentially all indicators. Even electrode-based membrane potential measurements are generally comparisons between two electrodes – one that is recording from within the cell and one that is a reference in the bath. The advantage of VF-FLIM is in the reproducibility and extensibility of the lifetime-voltage calibration. In stark contrast to calibrations that are obtained with other indicators (CAESR lifetime or di-8-ANEPPS excitation ratios; Figure 1—figure supplement 4 and Figure 1—figure supplement 5, Figure 1—source data 3), we demonstrate that the lifetime-voltage calibrations we record are consistent enough between cells such that calibrations on a subset of cells can be extended to other cells of the same line. This extensibility is relatively new; in the case of di-8-ANEPPS, the Loew laboratory showed that concurrent electrophysiology is necessary on each cell to compensate for the effects of membrane dipole potential (Zhang et al., 1998).

While collecting data for VF-FLIM, we tested the stability of the calibrations rigorously. Recordings in HEK293T cells in this manuscript are compiled data collected intermittently across 18 months, using many thaws of cells and a range of passage numbers. Over these 18 months, the instrument itself underwent a factory rebuild of the laser, multiple realignments of the optical table, and replacement of the photon counting card to a newer model. We have also made the same measurements on a separate TCSPC FLIM instrument (Zeiss LSM 880 confocal with FLIM electronics from a different supplier and a diode laser instead of the MaiTai Ti:Sapphire laser). On this other system, we found that the measured lifetime-V_mem_ calibration in HEK293T was almost identical (slope [this manuscript] = 3.50 ± 0.08 ps/mV; slope [other system] = 3.43 ± 0.08 ps/mV; 0 mV τ [this manuscript]=1.77 ± 0.02 ns; 0 mV τ [other system] = 1.75 ± 0.03 ns; mean ± SEM of n=17 cells for this manuscript, n=6 cells for the other system). So, we believe that the calibrations are stable and extensible enough to be useful without excessive re-testing with an electrode.

While some cell lines display differences in slope or 0 mV lifetime from other cell lines, the spread in 0 mV lifetimes is actually not cell type dependent. The variance in 0 mV lifetimes is statistically identical among cell lines (Levene’s test on the median for homoscedasticity: F(4,67) = 1.29, p = 0.28; Bartlett’s test: T = 3.76, p = 0.44). We were incorrect in stating this previously; the request for more explicit statistical analysis from Reviewer 4 in Question 6 below brought this to our attention. The higher inter-cell error in MCF-7s is partially the result of the lower sensitivity, which makes the V_mem_ equivalent noise higher. We appreciate this comment from the reviewer and have modified the Results section to enumerate differences between cell lines in a more statistically accurate way (subsection “Evaluation of VF-FLIM across Cell Lines and Culture Conditions”).

Yes, the τ_fl_-V_mem_ calibration would need to be established for each new cell type under study. While we think the need for calibration is clear throughout the text, we have added more discussion of the extensibility of the calibration (subsection “Resolution of VF-FLIM: Voltage, Space, and Time”).

Many images show blobs (Figure 1E and F, Figure 2C, Figure 3, Figure 4, Figure 5B). Sometimes the blobs seem voltage dependent sometimes not. I would presume that the blobs are the result of non-specific staining. This subject was not discussed. Were the presented images selected for relatively good membrane staining?

All images in the manuscript are representative. Enabling readers to assess the quality of data is of primary importance to us. To address this, in our original submission, we provided many images in both the main text and supporting info to orient readers to the type and quality of data that is generated with VF-FLIM. See, for example, Figure 2—figure supplement 1, Figure 3—figure supplement 1, Figure 3—figure supplement 2, Figure 4—figure supplement 1, Figure 4—figure supplement 3. In total, we include over 240 separate images, taken from greater than 70 separate samples.

We are not certain what is meant by the term blobs, so we address two potential aspects of this comment below.

First, there is small, extracellular debris in some fields of view (see for example the red punctum Figure 3—figure supplement 2C). This sparse lipophilic debris is inherent in cultured cell preparations. VF2.1.Cl stains these blobs, but we exclude these particles from the analysis based on the clear morphological difference between the debris and cells. These blobs therefore have no effect on the VF-FLIM technique or analysis.

Second, there is punctate fluorescence inside some cells. We believe this signal originates from internalized VoltageFluor. We generally have sufficient spatial resolution to exclude these puncta from the analysis. All region of interest identification was done on morphology alone and without knowledge of the lifetime data. Regardless, inclusion of this internal signal does not substantially affect the lifetime results. We addressed the effects of this small amount of internal signal in Author response image 1. In that analysis, we show that naively including all internal pixels in the HEK293T electrophysiology dataset does not have a large effect on the optical V_mem_ determinations.

In Figure 4A and Figure 5B different parts of the cell membrane appear to have different voltage responses and these response differences do not seem to be stochastic. This result does not fit with expected membrane voltage uniformity for small cells.

Yes, we agree that pixel to pixel differences in the lifetime are unlikely to reflect differences in V_mem_ in small, isolated, approximately spherical cells. We think that the lifetime differences across groups of adjacent cells such as those in Figure 4 and Figure 5 may reflect true voltage differences arising from differences in electrical coupling, which we indirectly saw by attempting to voltage clamp groups of cells (now Figure 2—figure supplement 3). Generally, though, pixel-to-pixel lifetime differences are probably not true voltage differences. We think the most likely source of this variability is noise in the fit of a biexponential decay model at each pixel. Instead, individual pixels of the image show higher random noise at each V_mem_; averages across multiple pixels produce more robust results. Indeed, lifetime determinations at individual pixels incorporate information from 20- to 100-fold fewer photons than the lifetime determinations for the ROI as a whole. We interpret lifetimes as a cellular average (see Figure 1—figure supplement 1), so we do not interpret pixel-to-pixel variability as V_mem_ in our analysis. In principle, we likely could reduce this pixel to pixel variability by collecting many more photons per image, but this would require longer acquisition times. We now mention heterogeneity in FLIM images in our discussion of the spatial resolution of VF-FLIM (subsection “Resolution of VF-FLIM: Voltage, Space, and Time”).

Drawbacks of the method are not discussed. The time resolution seems relatively slow. Will the method will be applicable to preparations with substantial light scattering? How will it work in three dimensional preparations? The differences from cell type to cell type will require calibration for each cell type and perhaps for each developmental age of each cell type.

We agree that a more complete analysis of the pros and cons of VF-FLIM is helpful to readers. In our original submission, we compare advantages and disadvantages of various voltage measurement approaches in Figure 1—source data 1. In our revised submission, we have expanded the Discussion section to elaborate upon other factors that can affect the FLIM measurement (subsection “Resolution of VF-FLIM: Voltage, Space, and Time”), as well as the acquisition time required.

Regarding more complex samples, lifetime imaging can be performed in 3D preparations and in preparations with substantial light scattering. Ryohei Yasuda’s lab has extensively employed FLIM in such contexts, for example to monitor CAMKII activity in dendritic spines in cultured hippocampal slices (Lee et al., 2009). FLIM has also been demonstrated in clinical diagnostic settings (e.g. Dysli et al., 2017; Sparks et al., 2015). So, in principle, FLIM is extensible to such systems, although we have not yet attempted to do this. As part of the combined review comment that we modify our wording to be more circumspect, we have changed wording throughout the text to more precisely refer to “cell lines” rather than “cell types” or “biological contexts,” making it clear that we have not yet explored VF-FLIM in systems other than immortalized cell culture.

There are a few considerations to be addressed before FLIM for absolute voltage imaging could be applied in complex preparations. The identification of signal from specific cells would require targeting of VoltageFluors, which we mention in subsection “Resolution of VF-FLIM: Voltage, Space, and Time” and “Epidermal Growth Factor Induces Vmem Signaling in A431 Cells”). Furthermore, there is the issue of changes in calibration between cell types.

We discuss the extensibility of the lifetime-V_mem_ calibration in detail in response to reviewer 4, above.

[Editors' note: further revisions were requested prior to acceptance, as described below.]

The manuscript has been improved but there are some remaining issues that need to be addressed before acceptance, as outlined below:Reviewer #2:[…]1) In comparing the ratiometric electrochromic dyes in the Introduction, they state: "Although they benefit from simpler loading procedures, signals from electrochromic styryl dyes display a strong dependence on local membrane properties other than transmembrane potential, reducing the accuracy of V_mem_ determinations (Gross et al., 1994; Montana et al., 1989; Zhang et al., 1998)". This is correct when examining different cell lines or cells in different states of differentiation, where the lipid composition, for example, can affect the ratio. But saying this in the Introduction appears to suggest that the new method that is about to be described does not have this problem; it likely does and even shows differences within the same cell line. And, actually, examination of Figure 2 in Zhang et al. (1998) shows that the cell to cell variation for the normalized ratiometric approach is remarkably small for the 40 cells examined. Figure 4 in Zhang et al. (1998) shows remarkably little variation in V_rest_ for undifferentiated neuroblastoma cells; the small variation in V_rest_ for differentiated cells, may be attributed to different degrees of differentiation.

In these lines, we did not mean to indicate that VoltageFluors are insensitive to membrane composition. We removed the reference to dipole potential sensitivity from this part of the Introduction, reserving it for the Discussion section where we talk about V_mem_ resolution (Gross, Bedlack, and Loew, 1994).

We now include an additional reference to illustrate the performance of normalized and nonnormalized ANEPPS ratios (Bullen and Saggau, 1999).

The text now reads (Introduction):

“While they benefit from simpler loading procedures, signals from electrochromic styryl dyes require normalization with an electrode on each cell of interest to determine absolute V_mem_ accurately (Bullen and Saggau, 1999; Montana et al., 1989; Zhang et al., 1998). As a result, ratiometric V_mem_ sensors cannot be used to optically quantify slow signals in the resting V_mem_, which may be on the order of tens of millivolts. Indeed, ratiometric V_mem_ probes are most commonly applied to detect – rather than quantify – fast changes in V_mem_ (Zhang et al., 1998), much like their single wavelength counterparts.”

Although the normalized ratiometric signal from di-8-ANEPPS does display low variability, we do not believe this is the most appropriate comparison to VF-FLIM (see our discussion of this in the manuscript in subsection “VoltageFluor Fluorescence Lifetime Varies Linearly with Membrane Potential**”**). While normalized signals are effective for detecting and quantifying changes in V_mem_ with high signal to noise, they cannot be used to quantify absolute V_mem_ without a point of reference. The use of electrode on each cell of interest (as in Zhang et al., 1998) improves the V_mem_ resolution, but the technique then suffers from many of the drawbacks of patch clamp electrophysiology. A key feature of VF-FLIM is that the calibration generated can be applied to many cells without an electrode, opening up the possibility of analyzing 1000s of cells (Figure 3) or V_mem_ dynamics accompanied by cellular movement (Figure 4 and Figure 5).

2) In discussing the influence of membrane dipole potential, the authors misunderstand some of the studies from my lab, subsection “Resolution of VF-FLIM: Voltage, Space, and Time”:"Relative to di-8-ANEPPS, where this effect was documented (Gross et al., 1994; Zhang et al., 1998), VF-FLIM displays less cell to cell variability, suggesting reduced dependence on the membrane dipole potential. The reason for this is unclear, as both sensors putatively detect V_mem_ from within the plasma membrane (Loew et al., 1979; Miller et al., 2012)." Our studies deliberately sought to establish the sensitivity of d-8-ANEPPS to dipole potential by systematically measuring ratios with different lipid compositions in lipid vesicles and by adding or depleting cholesterol in cell membranes. As a side benefit, these studies showed that membrane composition had to be considered when using dual wavelength ratio measurements to determine absolute V_mem_. Until the authors do a deliberate investigation of this effect on FLIM of their probes, I don't think they can say that FLIM of the VF dyes is less sensitive to membrane composition.

We agree; we did not evaluate the sensitivity of VF dyes to membrane composition or dipole potential. In the quoted lines above, we meant that VF-FLIM shows less cell to cell variability than di-8-ANEPPS does in HEK cells. Since we do not have direct evidence that the dipole potential is the reason for these differences, we have removed this text. We realize that the ability of di-8-ANEPPS to report dipole potential can be advantageous, and we did not mean to imply that this was a downside of the indicator. The text now reads as follows (subsection “Resolution of VF-FLIM: Voltage, Space, and Time”):

“Secondly, membrane composition and dipole potential can vary between cells and cell lines, changing the local environment of the fluorescent indicator (Wang, 2012; Brügger, 2014). Styryl dyes like di-8-ANEPPS can respond to changes in dipole potential (Gross et al., 1994; Zhang et al., 1998), and VF dyes may be similarly sensitive to dipole potential.”

Reviewer #4:[…]In the Abstract add the words "in culture" after the words "single cell resolution".

We have added “…in mammalian cell culture…” to the Abstract.

The line in question now reads, “To address this need, we developed a fluorescence lifetime-based approach (VF-FLIM) to visualize and optically quantify V_mem_ with single-cell resolution in mammalian cell culture.”

The first paragraph of the Introduction leads the reader to think that the paper is about signals that can be measured rapidly and can be measured in complex tissues even though the reported measurements have a time resolution that is ~four orders of magnitude slower than presently available from other methods and the measurements are only from single cells in culture. .

We removed the clause “At the tissue and organismal level…” from the final sentence of the first paragraph. Now there is no mention of tissues, allowing us to direct the attention of the reader to thinking about (1) the diversity of time scales over which voltage may be important and (2) the distinction between fast activity and the more gradual “resting” membrane potential signals which we go on to study in this work. We think this introduction now firmly places the focus on cellular studies.

Regarding time resolution, in the first round of revisions, we provided detailed information regarding the resolution that we achieved in these systems: in the Introduction, extensively detailed in the Materials and methods section, and Discussion section. To improve the clarity surrounding the temporal resolution, we have added additional emphasis on this throughout the text (see response to reviewer 4’s third point below).

The timing issue would be clearer for the reader if the table in subsection "Acquisition Time and Effective Pixel Size in Lifetime Data” was in the main body of the paper.

To ensure that this aspect of our experimental design is clear to the reader, we increased the visibility of the time resolution throughout the Introduction and Results sections, mentioning acquisition times when the associated data is presented, in addition to the tabulated imaging parameters in the supporting information. Changes are listed below.

a) In the Introduction where we first mention the V_mem_ resolution of VF-FLIM: “Using patch-clamp electrophysiology as a standard, we demonstrate that VF-FLIM reports absolute membrane potential in single trials and 10 to 23 mV accuracy (root mean square deviation, RMSD; 15 second acquisition), depending on the cell line.”

b) With the results of VF-FLIM calibration in HEK293T (subsection “VoltageFluor Fluorescence Lifetime Varies Linearly with Membrane Potential”): “We estimate that the resolution for tracking and quantifying voltage changes in a single HEK293T cell is 3.5 ± 0.4 mV […] whereas the resolution for single-trial determination of a particular HEK293T cell’s absolute V_mem_ is 19 mV […] within a 15 second bandwidth.”

c) With the VF-FLIM calibration in A431, CHO, MCF-7, and MDA-MB-231 cells (subsection “Evaluation of VF-FLIM across Cell Lines and Culture Conditions”): “For absolute V_mem_ determination of a single cell, we observed voltage resolutions ranging from 10 to 23 mV (inter-cell resolution, 15 second acquisition time, Figure 2—source data 1).”

d) With the EGFR dynamics data in Figure 4 (subsection “Membrane potential dynamics in epidermal growth factor signalling”): “We find that treatment of A431 cells with EGF results in a 15 mV hyperpolarization within 60-90 seconds in approximately 80% of cells (Figure 4A-C, Figure 4—figure supplement 1, Figure 4—figure supplement 2), followed by a slow return to baseline within 15 minutes (Figure 4D-F, Figure 4—figure supplement 3, 30 second acquisitions).”

The time resolution of VF-FLIM is not an invariant quantity. It depends on many factors, including the efficiency of the confocal light path and the desired level of V_mem_ accuracy. Faster TCSPC FLIM measurements could be (and have been) made, especially with improved equipment (see response to reviewer 4’s fifth point, below).

A discussion of the difficulties of applying this method to other applications should be added to the Discussion section.

We expanded our discussion for applying VF-FLIM to tissues. We now discuss the considerations of probe loading into specific cells of interest in subsection “Resolution of VF-FLIM: Voltage, Space, and Time”.

“When applying VF-FLIM to tissues, the cellular specificity of the VF stain becomes a consideration, as the VF2.1.Cl indicator used in this study labels all cell membranes efficiently.”

As part of the first round of revisions, we added discussion of difficulties and considerations for applying VF-FLIM to other applications, including the following:

1) Confounding factors such as temperature, viscosity, etc. (see subsection “Resolution of VF-FLIM: Voltage, Space, and Time”, quoted below):

“Additionally, fluorescence lifetime depends on certain environmental factors (e.g. temperature, viscosity, ionic strength) (Berezin and Achilefu, 2010), which may introduce variability. These parameters are usually determined by the biological system under study, and recalibration is important if they change dramatically in an experiment.”

2) Pixel to pixel variability in the lifetime images (subsection “Resolution of VF-FLIM: Voltage, Space, and Time”)

“Intriguingly, there are differences in lifetime within some cells in VF-FLIM images at the pixel to pixel level. In small, mostly spherical cells under voltage clamp, one would expect uniform membrane potential (Armstrong and Gilly, 1992), so these subcellular differences are most likely noise in the measurement. […] Lifetime estimates at each pixel are calculated from 20 to 100-fold fewer photons than the lifetime value for the entire ROI. These lower photon counts at the single pixel level produce V_mem_ estimates that are less precise than the V_mem_ estimate for the entire ROI.”

3) The need for calibration (subsection “Resolution of VF-FLIM: Voltage, Space, and Time”)

“One remaining challenge in expanding VF-FLIM to these areas is the requirement for an initial calibration with voltage clamp electrophysiology. Alternative ways to control V_mem_, such as ionophores or optogenetic actuators (Berndt et al., 2009), may prove useful in these systems.”

4) Slower acquisition speed than intensity-based imaging (subsection “Resolution of VF-FLIM: Voltage, Space, and Time”)

“[…] VF-FLIM sacrifices some of the temporal resolution of electrophysiology or intensity-based voltage imaging. VF-FLIM acquisition times are limited by the large numbers of photons needed per pixel in time-correlated single photon counting (see Materials and methods section). As a result, VF-FLIM in its current implementation can track V_mem_ events lasting longer than a few seconds. For “resting” membrane potential or V_mem_ dynamics associated with cell growth or differentiation, this temporal resolution is likely sufficient. Nevertheless, in the future, we envision allying VF-FLIM with recently developed, faster lifetime imaging technology to enable optical quantification of more rapid V_mem_ responses (Gao et al., 2014; Raspe et al., 2016).”

5) The need for improved voltage sensitivity/voltage resolution, (subsection “Epidermal Growth Factor Induces V_mem_ Signaling in A431 Cells”)

“Future improvements to the voltage resolution could be made by use of more sensitive indicators, which may exhibit larger changes in fluorescence lifetime (Woodford et al., 2015). VF-FLIM can be further expanded to include the entire color palette of PeT-based voltage indicators (Deal et al., 2016; Huang et al., 2015) […]”

6) As well as for targeting for use in complex tissues (subsection “Epidermal Growth Factor Induces Vmem Signaling in A431 Cells”)

“VF-FLIM can be […] allied with targeting methods to probe absolute membrane potential in heterogeneous cellular populations (Grenier et al., 2019; Liu et al., 2017) […]”

We’re happy to elaborate on other specific applications, as necessary.

The Discussion section notes that faster apparatus is available. How much faster? One order?

We think that two orders of magnitude improvement can be readily achieved.

The simplest upgrade is an improvement to the optics and photon economy of the current confocal microscope, which is 20 years old. We recently evaluated a new time-domain lifetime microscope, and we find that we are able to acquire lifetime data in beating cardiomyocytes derived from human induced pluripotent stem cells at a rate of approximately 8 Hz (>200-fold improvement over our current 0.033 Hz rate). After an exhaustive review by the NIH, we were recently informed that our core facility’s S10 application was funded, so the real possibility of using this microscope looms on the horizon.

In addition to upgrading to a more modern confocal microscopy equipped with TCSPC, there are two new instrumentation schemes that we think, in principle, should allow faster acquisition, which we refer to in subsection “Resolution of VF-FLIM: Voltage, Space, and Time” and subsection “Epidermal Growth Factor Induces Vmem Signaling in A431 Cells”, quoted below:

Subsection “Resolution of VF-FLIM: Voltage, Space, and Time”: “Nevertheless, in the future, we envision allying VF-FLIM with recently developed, faster lifetime imaging technology to enable optical quantification of more rapid V_mem_ responses (Gao et al., 2014; Raspe et al., 2016).”

Subsection “Epidermal Growth Factor Induces Vmem Signaling in A431 Cells”: “VF-FLIM can be further expanded to include the entire color palette of PeT-based voltage indicators (Deal et al., 2016; Huang et al., 2015), allied with targeting methods to probe absolute membrane potential in heterogeneous cellular populations (Grenier et al., 2019; Liu et al., 2017), and coupled to highspeed imaging techniques for optical quantification of fast voltage events (Gao et al., 2014; Raspe et al., 2016).”

In Raspe et al. (2016), the authors show that it is possible to track Ca^2+^ transients with Oregon Green BAPTA 1 in HeLa cells at a rate of 6 Hz and in HL-1 cardiomyocytes at a rate of 20 Hz. We have not evaluated the performance of VF dyes with this camera system. An acquisition rate of 6 Hz (~170 ms acquisition time) would be a ~180-fold increase over our current acquisition rate for the EGF signaling data, which was 0.033 Hz (30 second acquisition time). A 20 Hz acquisition rate would represent ~600-fold increase.

In Gao et al. (2014), the authors combine a streak camera with compressed sensing to achieve framerates of up to 100 GHz. They estimate that, in this current iteration, a frame size of 150 x 500 pixels could be achieved. In principle, this approach could provide exceptionally fast lifetime imaging.

Please discuss the possible explanations for "between regions" in the following sentence in the Discussion section: "Intriguingly, there are differences in lifetime within some cells in VF-FLIM images, both at the pixel to pixel level and between regions of the cell membrane."

We think that the two most likely explanations for subcellular differences in lifetime are fit noise and membrane composition, which we discussed in the first round of revisions of the manuscript immediately after the aforementioned line. We have changed the wording of this paragraph slightly to clarify this point (see below).

We did not mean to imply a mechanistic distinction between the “pixel to pixel” noise and the “between regions” noise. As such, we have edited the sentence you mention to the following (subsection “Resolution of VF-FLIM: Voltage, Space, and Time”):

“Intriguingly, there are differences in lifetime within some cells in VF-FLIM images at the pixel to pixel level.”

We also have reworded the final sentence of this paragraph to more accurately represent the potential effects of membrane composition difference (creating subcellular differences rather than simply *pixel to pixel* ones; subsection “Resolution of VF-FLIM: Voltage, Space, and Time”).

“We also cannot fully rule out an alternative explanation that the observed subcellular variability is the result of local differences in membrane composition (Gross et al., 1994).”